# Mitigating a TDP-43 proteinopathy by targeting ataxin-2 using RNA-targeting CRISPR effector proteins

M. Alejandra Zeballos C. ®[1], Hayden J. Moore[1], Tyler J. Smith[1], Jackson E. Powell[1], Najah S. Ahsan[1], Sijia Zhang[1] & Thomas Gaj ®[1,2] ✉

The TDP-43 proteinopathies, which include amyotrophic lateral sclerosis and frontotemporal dementia, are a devastating group of neurodegenerative disorders that are characterized by the mislocalization and aggregation of TDP-43. Here we demonstrate that RNA-targeting CRISPR effector proteins, a programmable class of gene silencing agents that includes the Cas13 family of enzymes and Cas7–11, can be used to mitigate TDP-43 pathology when programmed to target ataxin-2, a modifier of TDP-43-associated toxicity. In addition to inhibiting the aggregation and transit of TDP-43 to stress granules, we find that the in vivo delivery of an ataxin-2-targeting Cas13 system to a mouse model of TDP-43 proteinopathy improved functional deficits, extended survival, and reduced the severity of neuropathological hallmarks. Further, we benchmark RNA-targeting CRISPR platforms against ataxin-2 and find that high-fidelity forms of Cas13 possess improved transcriptome-wide specificity compared to Cas7–11 and a first-generation effector. Our results demonstrate the potential of CRISPR technology for TDP-43 proteinopathies.

The mislocalization and aggregation of TDP-43, a nuclear DNA/RNA binding protein involved in the regulation of transcription[1], is a cardinal feature of a group of neurodegenerative disorders known as the TDP-43 proteinopathies[1–6]. This includes nearly all cases of amyotrophic lateral sclerosis (ALS)[2,4], a disorder characterized by the deterioration of motor neurons in the brain and spinal cord[7], as well as the most common subtype of frontotemporal dementia (FTD)[2,4], a syndrome involving the degeneration of the frontal and/or temporal lobes of the brain. Though the exact pathological role of TDP-43 in ALS-FTD and other TDP-43 proteinopathies remains incompletely understood, it's believed that its depletion from the nucleus and/or its abnormal deposition into cytoplasmic inclusions may give rise to gain and/or loss of functions that drive its toxicity[1,8–10], a hypothesis supported by the discovery that its mutation is also associated with disease[11–14].

However, despite its central role in neurodegeneration, numerous lines of evidence suggest that TDP-43 may not be an appropriate target for therapeutic silencing, as it plays a key role in numerous cellular processes[15] and its depletion from cells can induce side effects[16–19]. Instead, various attempts have been undertaken to identify modifier(s) whose targeting can attenuate the apparent toxic effects induced by aberrant TDP-43 without affecting its underlying abundance[20–24]. One such modifier is ataxin-2[20,25,26], a polyglutamine (polyQ)-containing RNA binding protein thought to be involved in regulating mRNA translation[27] and stress granule (SG) assembly[26,28,29] that, when mutated to carry an intermediate-length polyQ expansion[20,30,31], also emerges as a risk factor for ALS[20,30,32–35]. Importantly, the potential for ataxin-2 as a therapeutic target is supported by several lines of evidence, including that its downregulation in yeast and flies decreases TDP-43-associated toxicity[20] and that its genetic ablation in a rodent model of TDP-43 proteinopathy can mitigate pathological features without affecting TDP-43 expression[26]. Thus, strategies capable of modulating the expression of ataxin-2 may hold potential for ALS-FTD and other TDP-43 proteinopathies.

[1]Department of Bioengineering, University of Illinois Urbana-Champaign, Urbana, IL 61801, USA. [2]Carl R. Woese Institute for Genomic Biology, University of Illinois Urbana-Champaign, Urbana, IL 61801, USA. ✉e-mail: gaj@illinois.edu

In addition to traditional gene silencing modalities, CRISPR technologies have emerged as effective platforms capable of perturbing the expression of a target gene[36–38], including potentially ataxin-2. Among the CRISPR-based platforms that can be used for this goal are RNA-targeting CRISPR effector proteins[39–45], which rely on an engineered CRISPR RNA (crRNA) molecule to recognize a target RNA sequence and facilitate its degradation by the CRISPR protein. This continuously expanding group of effector proteins includes: (i) the Cas13 family of enzymes[39,41–44], which comprise four subtypes (a, b, c, and d) that each cleave RNA via their conserved higher eukaryotes and prokaryotes nucleotide-binding (HEPN) domains and (ii) Cas7–11[46], a single-protein effector that likely originated from the fusion of a putative Cas11 domain and multiple Cas7 subunits[46]. Given their programmability and their capacity to knock down the expression of a target gene in mammalian cells[39,44], we hypothesized that RNA-targeting CRISPR effectors could be harnessed to target ataxin-2 for the goal of attenuating a TDP-43 proteinopathy.

Here we show that, when programmed to target ataxin-2, RNA-targeting CRISPR effectors are capable of mitigating TDP-43 pathology. In addition to inhibiting the aggregation and transit of the TDP-43 protein to SGs, highly dynamic intracellular accumulations of protein and RNA that are thought to play a role in ALS-FTD[29,47,48], we demonstrate that this class of CRISPR-based proteins can be harnessed to slow disease progression in vivo. Specifically, we find that delivering an ataxin-2-targeting Cas13 system to a mouse model of TDP-43 proteinopathy could decrease ataxin-2 mRNA in the brain and spinal cord, an outcome that we show improved functional deficits, increased survival, and reduced the severity of several neuropathological hallmarks. Further, we benchmark recently emerged RNA-targeting CRISPR effector platforms against ataxin-2 and find that high-fidelity forms of the Cas13 protein possess improved targeting specificity compared to Cas7–11 and a first-generation Cas13 protein. Our results demonstrate the potential of CRISPR technologies for TDP-43 proteinopathies and reinforce the ability of ataxin-2 to modify TDP-43 toxicity.

## Results

### Targeting ataxin-2 with RfxCas13d

Because of its ability to influence TDP-43-associated toxicity[20,25,26], ataxin-2 has emerged as a potentially broadly applicable target for ALS-FTD, as TDP-43 pathology is observed in ~97% of ALS cases[1,4,5] and ~45% FTD occurrences[1,2,4]. Given this, we sought to determine if RNA-targeting CRISPR effector proteins could be used to lower the expression of ataxin-2 to mitigate TDP-43-mediated toxicity (Fig. 1a). As overexpression of the human TDP-43 protein in mice can result in a phenotype reminiscent of ALS that can be attenuated by the knockout of its ataxin-2 gene[26], we initially sought to target the mouse ataxin-2 (mATXN2) transcript to ensure that this strategy could be assessed in a validated rodent model.

To target the mATXN2 mRNA, we first used the Cas13d nuclease from *Ruminococcus flavefaciens* XPD3002 (RfxCas13d)[44], a programmable RNA-targeting CRISPR effector protein that we[49] and others[50,51] have demonstrated can silence target gene expression in the nervous system. To facilitate the identification of crRNAs for mATXN2 in a relatively high-throughput manner, we created a reporter plasmid carrying the mATXN2 protein-coding sequence fused to an enhanced green fluorescence protein (EGFP) variant via a self-cleaving T2A peptide, which links mATXN2 expression to EGFP fluorescence, thereby enabling us to assess RfxCas13d-mediated targeting by flow cytometry. We then designed ten crRNAs to target mATXN2, focusing on the region corresponding to exons 5 to 15 (Fig. 1b and Supplementary Fig. 1). As computational methods capable of predicting active crRNAs for RfxCas13d were not available at the time we initiated this study, we based this targeting strategy on our prior results[49], which indicated that crRNAs designed to bind a region equivalent to this window could target their transcript efficiently.

Following the design of the crRNAs, we transfected human embryonic kidney (HEK) 293 T cells with expression vectors encoding RfxCas13d and a mATXN2-targeting crRNA alongside the mATXN2-T2A-EGFP-encoding plasmid. Based on flow cytometry, we found that

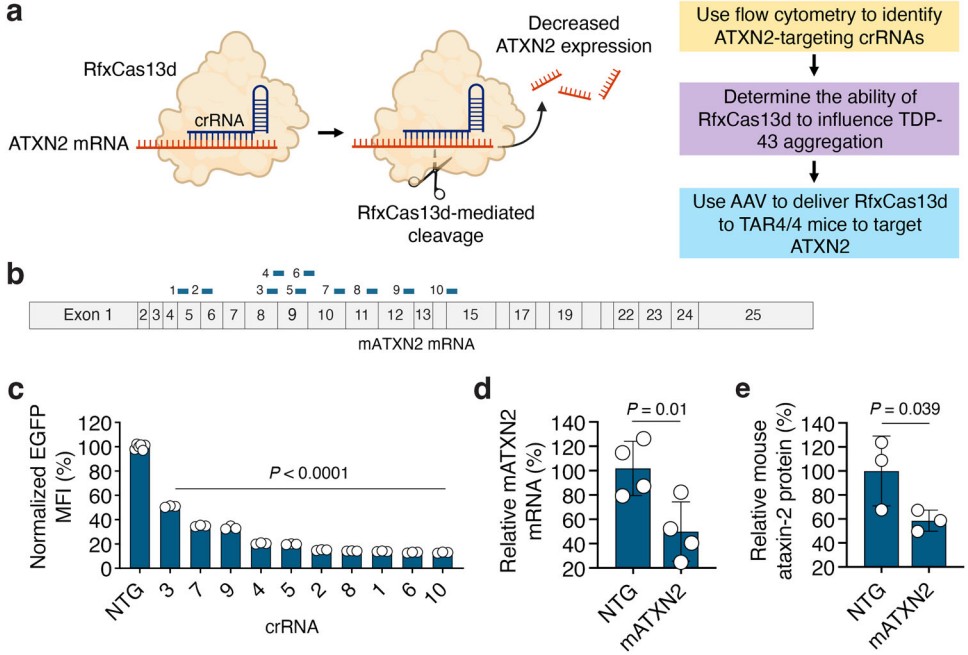

**Fig. 1 | Programming RfxCas13d to target ataxin-2. a** Overview of the strategy for targeting ATXN2 using RfxCas13d. **b** Schematic of the mature mouse ataxin-2 (mATXN2) mRNA and location of the crRNA (CRISPR RNA) binding sites (blue bars). **c** EGFP mean fluorescence intensity (MFI) in HEK293T cells transfected with mATXN2-T2A-EGFP alongside RfxCas13d and each mATXN2-targeting crRNA. Data are normalized to MFI from cells transfected with mATXN2-T2A-EGFP, RfxCas13d, and a non-targeted (NTG) crRNA ($n = 3$). **d**, **e** Relative (**d**) mATXN2 mRNA and (**e**) ataxin-2 protein from Neuro2A cells transfected with RfxCas13d and crRNA-10 at 48 hr post-transfection. Data are normalized to cells transfected with RfxCas13d and a NTG crRNA ($n = 4$ for mRNA, $n = 3$ for protein). All data points are biologically independent samples. Values represent means and error bars indicate SD. Data were compared using a one-tailed unpaired $t$-test, with the exact $P$-values shown.

all ten of the crRNAs decreased EGFP fluorescence by >50% compared to a non-targeted crRNA at 48 hr post-transfection ($P < 0.0001$; Fig. 1c and Supplementary Fig. 2), with the most active crRNA, the exon 15-targeting crRNA-10, found to decrease EGFP fluorescence by ~87% ($P < 0.0001$; Fig. 1c). Conveniently, we also found that crRNA-6, a crRNA whose target in mATXN2 is perfectly homologous to the equivalent site in human ATXN2 (hATXN2), also substantially decreased EGFP fluorescence ($P < 0.0001$; Fig. 1c), indicating its potential as a platform for targeting the human ortholog.

We next evaluated if RfxCas13d could target the endogenous mATXN2 transcript and if its action in cells induced collateral effects, as RfxCas13d possesses an innate ability to *trans*-cleave non-target RNAs after engaging with its target substrate, an outcome that can lead to the non-specific degradation of bystander RNAs[52–56].

To answer this first question, we transfected Neuro2A cells, a mouse neuroblastoma cell line, with RfxCas13d and crRNA-10, the most efficient mATXN2-targeting crRNA identified from our initial screen. According to qPCR, which we used to measure mATXN2 mRNA at 48 hr post-transfection, we found that RfxCas13d decreased mATXN2 mRNA by ~50% compared to cells transfected with a non-targeted crRNA ($P < 0.05$; Fig. 1d). A subsequent western blot further revealed that RfxCas13d decreased the mouse ataxin-2 protein in Neuro2A cells by ~42% at the same time-point ($P < 0.05$; Fig. 1e and Supplementary Fig. 3).

To measure if collateral effects were induced by targeting the endogenous mouse ataxin-2 transcript, we transfected Neuro2A cells with RfxCas13d and crRNA-10 alongside a mCherry-encoding surrogate reporter, whose fluorescence was used as a proxy for collateral activity. Because collateral *trans*-cleavage by RfxCas13d correlates with the expression profile of the target transcript[52,55], mCherry fluorescence was expected to decrease with increased collateral effects.

Relative to cells transfected with mCherry alongside RfxCas13d and a non-targeted crRNA, we measured no difference in mCherry fluorescence intensity in cells transfected with RfxCas13d and crRNA-10 ($P > 0.05$; Supplementary Fig. 4). By contrast, we observed a ~17% decrease ($P < 0.01$) in mCherry fluorescence in cells transfected with RfxCas13d and a crRNA-targeting mouse DKK1 mRNA, a strongly expressed transcript whose targeting we predicted would induce collateral *trans*-cleavage (Supplementary Fig. 4). These latter measurements confirm the ability of this method to detect collateral effects in cells.

Thus, we find that RfxCas13d can be programmed to target the endogenous mATXN2 mRNA and that its action in Neuro2A cells did not induce measurable collateral effects according to a fluorescence-based reporter assay.

## RfxCas13d targeting can attenuate the formation of TDP-43 aggregates

In addition to its role in coordinating protein translation, the ataxin-2 protein can regulate the formation of SGs[26,28,29], highly dynamic cytoplasmic structures that form in response to cellular stress and are implicated in various neurodegenerative disorders[29,47,48]. As knocking down ataxin-2 can reduce the susceptibility of TDP-43 to localize to SGs[26], we determined if targeting ataxin-2 by RfxCas13d could also inhibit the recruitment of TDP-43 to these structures and decrease its propensity to form aggregates[26,28,29,57–59].

Because TDP-43 does not strongly localize with SGs in Neuro2A cells[60], we instead evaluated the ability of RfxCas13d to inhibit TDP-43 aggregation in HEK293T cells which, when treated with the osmotic and oxidative stressor sorbitol, can form TDP-43⁺ SGs[29,61,62]. To enable this experiment (Fig. 2a), we first determined the ability of RfxCas13d to target hATXN2 in HEK293T cells, a cell line whose ATXN2 gene carries 23 CAG repeats, a number below the 30–33 repeats that have emerged as a susceptibility factor for ALS[20,30,31]. Following their transfection with RfxCas13d and crRNA-6, the crRNA whose target in the mATXN2 mRNA matches the equivalent site in the human ortholog, we

used qPCR and western blot to measure the relative abundance of the hATXN2 mRNA and the ataxin-2 protein, respectively. From these measurements, we found that RfxCas13d decreased both products in HEK293T cells by ~40% at 48 hr post-transfection ($P < 0.05$ for both; Fig. 2b, c and Supplementary Fig. 5).

To determine whether targeting the endogenous human ATXN2 transcript induced collateral effects, we next transfected HEK293T cells with RfxCas13d and crRNA-6 alongside an mCherry-encoding surrogate reporter, whose fluorescence was again used as a proxy for collateral *trans*-cleavage. Similar to our findings from Neuro2A cells, we measured no difference in mCherry fluorescence intensity in cells transfected with RfxCas13d and crRNA-6 versus the control ($P > 0.05$; Supplementary Fig. 6), though cells transfected with RfxCas13d and a validated crRNA for RPL4[63], a strongly expressed transcript whose targeting by RfxCas13d can induce collateral effects[63], were observed to have a ~50% reduction in mCherry fluorescence ($P < 0.0005$; Supplementary Fig. 6). These results demonstrate that RfxCas13d can target ATXN2 in a human cell line without inducing a significant degree of collateral effects via a fluorescence-based reporter assay.

We next determined if targeting hATXN2 by RfxCas13d could affect TDP-43 aggregation. HEK293T cells were therefore transfected with RfxCas13d and crRNA-6, the hATXN2-targeting crRNA, and exposed to 0.4 M of sorbitol for 1 hr at 48 hr post-transfection (Fig. 2a), the same time-point used to measure that RfxCas13d decreased the ataxin-2 protein by ~40% (Fig. 2c and Supplementary Fig. 5). Through an immunofluorescence-based analysis that utilized CellProfiler[64] to identify TDP-43⁺ aggregates, we measured that HEK293T cells transfected with RfxCas13d and crRNA-6 had a ~42% decrease in the relative number of TDP-43-immunoreactive inclusions per cell compared to the controls, which were transfected with RfxCas13d and a non-targeted crRNA ($P < 0.005$; Fig. 2d, e). Further, when paired with crRNA-6, we found that RfxCas13d decreased the overall percentage of cells with detectable TDP-43 inclusions by ~28% ($P < 0.05$; Fig. 2d, f) and that, for cells with detectable TDP-43 inclusions, it reduced the average size of those inclusions by ~47% ($P < 0.01$; Fig. 2d, g). Critically, we also found that RfxCas13d decreased the number of cells with foci positive for both TDP-43 and the SG marker G3BPl[65,66] (Ras-GAP SH3-domain-binding protein 1) by ~43% ($P < 0.05$; Fig. 2d, h), providing evidence that RfxCas13d-mediated targeting of hATXN2 decreased the transit of TDP-43 to SGs.

We next determined whether targeting hATXN2 could influence the aggregation of TDP-43 protein variants that have been linked to ALS[11]. To this end, HEK293T cells were transfected with RfxCas13d and crRNA-6 alongside one of three reporter plasmids each expressing a distinct TDP-43 variant with a propensity to misfold and aggregate[67,68]: TDP-43^A315T, TDP-43^G298S or TDP-43^M337V. To enable the microscopic visualization of TDP-43 aggregation, each variant was fused to the yellow fluorescent protein (YFP). Notably, these three TDP-43 mutations are among the five most common ALS-linked mutations in TDP-43, with each mutation located outside of the RNA recognition motif (RRM) domain that is believed to mediate an RNA-dependent association with the ataxin-2 protein[20]. Thus, based on these insights and prior studies[20,23], we expected each variant to retain its ability to interact with ataxin-2.

According to our immunofluorescent analysis, which was conducted at 48 h post-transfection, we found that RfxCas13d decreased the size of the YFP-TDP-43 inclusions, though the extent varied based on the mutation. In the case of TDP-43^A315T, RfxCas13d decreased the average size of the inclusions by ~41% ($P < 0.01$; Supplementary Fig. 7) and reduced the average number of these inclusions per transfected cell by ~45% ($P < 0.01$; Supplementary Fig. 7) while, for TDP-43^G298S and TDP-43^M337V, RfxCas13d decreased the average size of inclusions by only ~15% ($P < 0.05$ for both; Supplementary Fig. 7).

To investigate this difference further, we analyzed hATXN2 expression in cells transfected with each TDP-43-YFP variant, finding no difference in the relative abundance of hATXN2 mRNA in cells

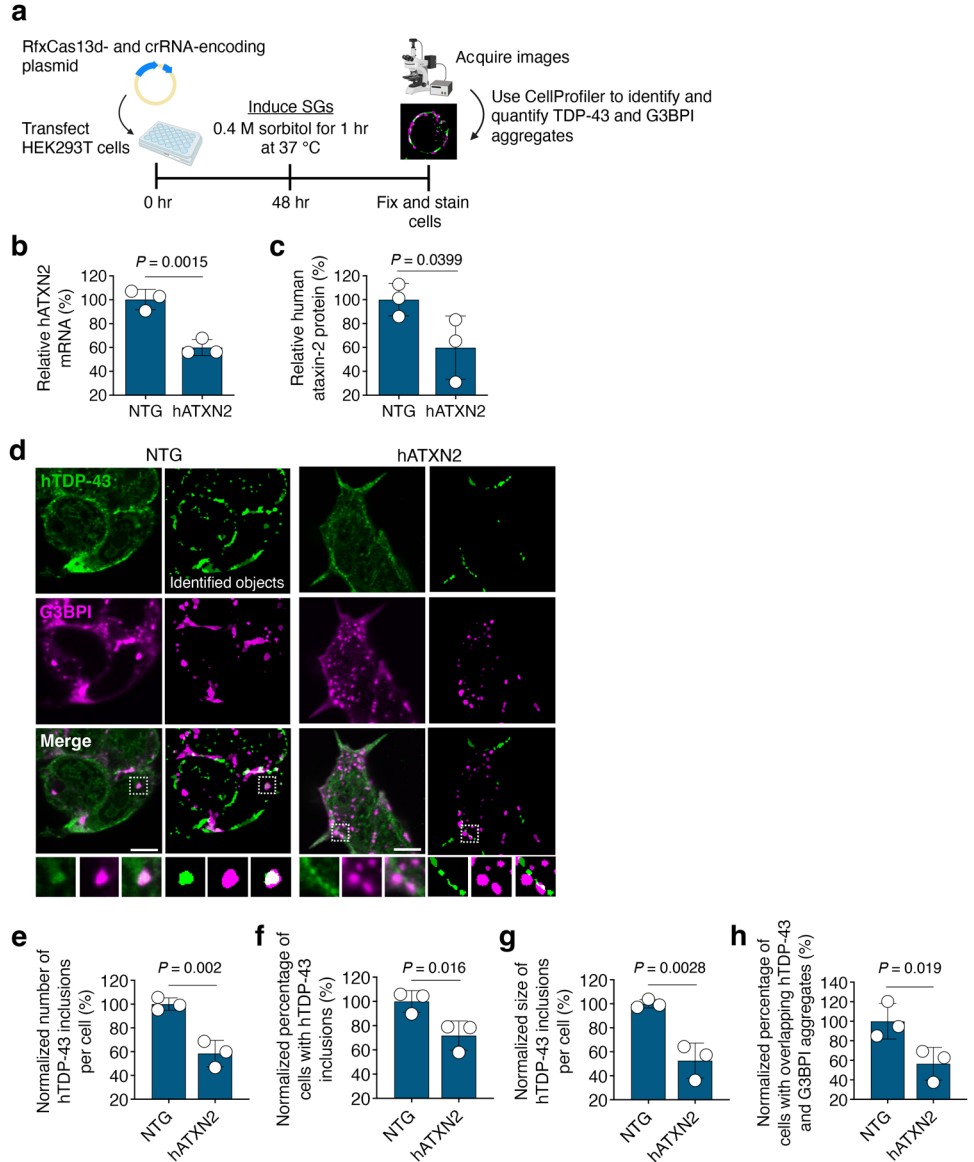

**Fig. 2 | RfxCas13d targeting reduces TDP-43 aggregation. a** Overview of the workflow for analyzing TDP-43 aggregation. **b, c** Relative (**b**) human ataxin-2 (hATXN2) mRNA and (**c**) ataxin-2 protein from HEK293T cells transfected with RfxCas13d and the hATXN2-targeting crRNA-6. Data are normalized to cells transfected with RfxCas13d and a non-targeted (NTG) crRNA ($n = 3$). **d** Representative immunofluorescent staining of HEK293T cells 48 hr after transfection with RfxCas13d and crRNA-6 and after treatment with sorbitol. CellProfiler was used to quantify sorbitol-induced TDP-43- and G3BPl-immunoreactive aggregates. A mask portraying the identified objects, with TDP-43 foci in green and G3BPl foci in magenta, is shown. Overlapping TDP-43$^+$ foci and G3BPl$^+$ foci were identified using the Relate Objects module, as depicted. >300 cells were analyzed per replicate.

G3BPl, the stress granule (SG) marker Ras-GAP SH3-domain-binding protein 1. Scale bar, 5 μm. Images were captured using identical exposure conditions. **e–h** Quantification of (e) the number of TDP-43$^+$ foci per individual cell; (**f**) the percentage of cells expressing one or more TDP-43$^+$ inclusions; (**g**) the area occupied by TDP-43$^+$ foci divided by the number of cells analyzed per image; (**h**) the percentage of cells identified as positive for both G3BPl and TDP-43 inclusions. All data are normalized to cells transfected with RfxCas13d and a NTG crRNA. All data were acquired at 48 hr post-transfection ($n = 3$). All data points are biologically independent samples. Values represent means and error bars indicate SD. Data were compared using a one-tailed unpaired $t$-test, with the exact $P$-values shown.

transfected with TDP-43$^{A315T}$, TDP-43$^{G298S}$ or TDP-43$^{M337V}$ versus the controls ($P > 0.05$ for all; Supplementary Fig. 8a), however, we did observe that inclusions for TDP-43$^{G298S}$ and TDP-43$^{M337V}$, which were less effectively reduced by RfxCas13d than TDP-43$^{A315T}$, were both ~2.5-fold larger in size than those for TDP-43$^{A315T}$ ($P < 0.001$ for both; Supplementary Fig. 8b), indicating that the size of the inclusions themselves could potentially affect the efficacy of this approach.

Finally, to rule out the possibility that the reduction in YFP-TDP-43$^+$ inclusions was attributable to collateral *trans*-cleavage of the YFP-TDP-43 transcript by RfxCas13d, we co-transfected HEK293T cells with RfxCas13d and crRNA-6 in the presence of the TDP-43-YFP reporter and an mCherry-encoding reporter. Flow cytometry revealed no

difference in mCherry or YFP fluorescence intensity in cells transfected with RfxCas13d with crRNA-6 versus the controls ($P > 0.05$ for both; Supplementary Fig. 9), indicating that the reduction in the TDP-43 inclusions was likely not attributable to collateral effects.

Altogether, these results demonstrate that RfxCas13d-mediated targeting of hATXN2 can influence the aggregation of TDP-43, including TDP-43 variants linked to ALS.

### Targeting ATXN2 in vivo by RfxCas13d can reduce a TDP-43 pathology

We next sought to determine the therapeutic benefit of targeting ATXN2 by RfxCas13d in a transgenic mouse model of a TDP-43

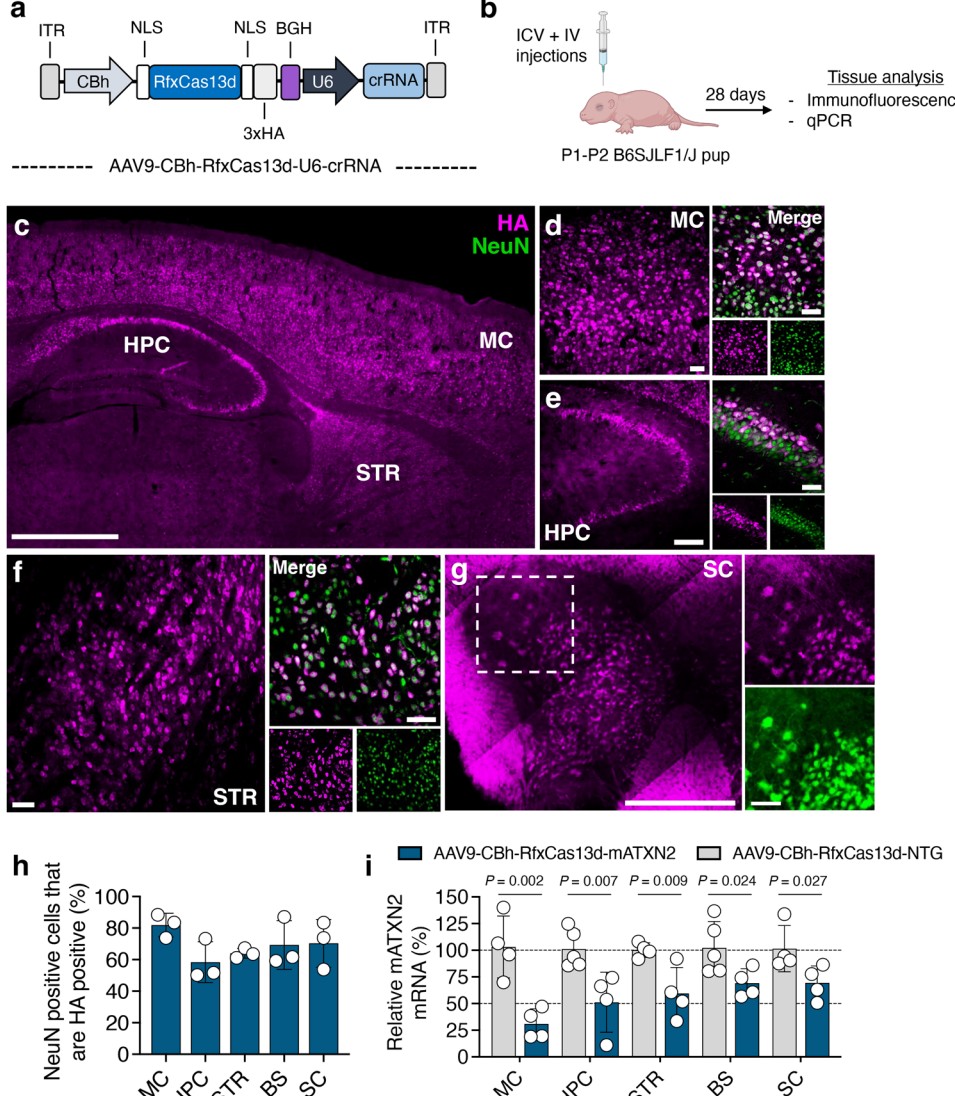

**Fig. 3 | RfxCas13d reduces ATXN2 mRNA in the brain and spinal cord.**
**a** Schematic of the AAV vector encoding RfxCas13d and its crRNA. ITR, inverted terminal repeat; CBh, chicken β-actin promoter; NLS, nuclear localization signal; BGH, bovine growth hormone polyadenylation signal. **b** Overview of the experimental plan. ICV, intracerebroventricular; IV, intravenous. **c–g** Representative immunofluorescence staining of the **c** dorsal sagittal section of the brain (scale bar, 1 mm), **d** motor cortex (MC; scale bar, 50 μm), **e** hippocampus [HPC; scale bar, (left) 100 μm, (right) 50 μm], **f** striatum (STR; scale bar, 50 μm), **g** spinal cord [SC; scale bar, (left) 500 μm, (right) 100 μm]. **c, g** Images were acquired by tile-scanning using an AxioScan.Z1 (Carl R. Woese Institute for Genomic Biology, Univ. Illinois) and subsequently stitched together to display a larger view. **h** Quantification of the percentage of NeuN⁺ cells positive for the HA epitope tag. Values represent means and error bars indicate SD (*n* = 3). >104 NeuN⁺ cells counted were per replicate.

**i** Relative mATXN2 mRNA in B6SJLF1/J mice injected with AAV9-CBh-RfxCas13d-mATXN2. Data are normalized to relative mATXN2 mRNA from B6SJLF1/J mice injected with AAV9-CBh-RfxCas13d-NTG. Values represent means and error bars indicate SD (*n* = 4 for the MC; *n* = 5 for NTG and *n* = 4 for mATXN2 in the HPC, *n* = 4 for the STR, *n* = 5 for NTG and *n* = 4 for mATXN2 in the BS; *n* = 4 for the SC). Data were compared using a one-tailed unpaired *t*-test, with the exact *P*-values shown. All analyses were conducted four weeks post-injection. All data points are biologically independent samples.

---

proteinopathy, specifically homozygous TAR4/4 mice, which carry the wild-type human TDP-43 gene under the control of a murine Thy-1 promoter[69], which preferentially expresses in neurons. TAR4/4 mice develop a quadriplegia phenotype reminiscent of ALS and have an average lifespan of ~24 days[69], which can necessitate early intervention.

To deliver RfxCas13d in vivo, we used adeno-associated virus serotype 9 (AAV9) which, when administrated to neonatal mice via intravenous (IV)[70] and intracerebroventricular (ICV)[71] injections, can access neurons in the brain and spinal cord. However, prior to delivering RfxCas13d to TAR4/4 mice, we administered it to wild-type animals to evaluate its distribution and targeting in a non-neurodegenerative background. Postnatal day 1-2 (P1-P2) B6SJLF1/J mice were therefore infused with an AAV9 vector encoding RfxCas13d,

driven by a CBh promoter, with either crRNA-10, the mATXN2-targeting crRNA identified from our initial screen (AAV9-CBh-RfxCas13d-mATXN2) or a non-targeted crRNA (AAV9-CBh-RfxCas13d-NTG; Fig. 3a). Because combining IV and ICV injections is an effective strategy for enhancing AAV9 delivery to the nervous system of neonatal mice[72–74], AAV was administered via both the facial vein (7 × 10¹¹ vector genomes [vg] per pup) and the lateral ventricles (8.4 × 10¹⁰ vg per pup; Fig. 3b).

Using its hemagglutinin epitope (HA) tag for detection, we observed at four weeks post-injection that RfxCas13d was present throughout the brain, including in the motor, somatosensory, and visual cortices (MC, SSC and VC respectively), the brainstem (BS), the striatum (STR), the hippocampus (HPC), the cerebellum (CBM), and

the spinal cord (SC; Fig. 3c–g and Supplementary Fig. 10a–g), with its expression largely confined to NeuN[+] cells (Fig. 3d–h Supplementary Fig. 10a–g and Supplementary Fig. 11).

To evaluate targeting, we used qPCR to measure the relative abundance of mATXN2 mRNA in several of the most strongly transduced areas, including the MC, the HPC, the STR, the BS, the SC, and the VC. Compared to animals injected with the non-targeted vector, mice injected with AAV9-CBh-RfxCas13d-mATXN2 had a ~70%, a ~50%, a ~40%, a ~30%, a ~30% and a ~50% decrease in relative mATXN2 mRNA in the MC, the HPC, the STR, the BS, the SC, and the VC, respectively ($P < 0.05$ for the BS and SC; $P < 0.01$ for MC, HPC, STR; $P < 0.001$ for VC; Fig. 3i and Supplementary Fig. 10h), indicating that RfxCas13d could suppress mATXN2 expression in vivo.

We next determined whether targeting mATXN2 induced adverse effects. More specifically, we analyzed neuronal viability and neuroinflammation in wild-type B6SJLF1/J mice injected with AAV9-CBh-RfxCas13d-mATXN2 or AAV9-CBh-RfxCas13d-NTG. Within both the MC and the BS, we observed no difference in the number of NeuN[+] cells or the number of GFAP[+] and Iba1[+] cells in mice injected with AAV9-CBh-RfxCas13d-mATXN2 versus the control vector ($P > 0.05$ for all; Supplementary Fig. 12). Though these results indicate that lowering mATXN2 did not affect neuronal viability or increase the proliferation of astrocytes or microglia for at least the first four-weeks post-injection in B6SJLF1/J mice, a more comprehensive and longer-term study is needed to conclusively determine the tolerability of this approach.

To determine whether RfxCas13d could influence TDP-43 pathology in vivo, we injected TAR4/4 mice at P1-P2 with AAV9-CBh-RfxCas13d-mATXN2 or AAV9-CBh-RfxCas13d-NTG using the same doses and dual administration strategy as implemented for B6SJLF1/J mice (Fig. 4a). In addition to abnormal gait and severe tremors, which typically begin manifesting at P14[69], TAR4/4 mice display a rapid and progressive decline in motor function and develop increasingly severe kyphosis[69], defined as a hunched posture resulting from a curved spine. We therefore monitored these phenotypes, as well as weight and lifespan, which are also adversely affected in TAR4/4 mice[69]. Notably, the murine ATXN2 gene carries only a single CAG[75] and therefore does not possess a polyQ expansion in its ataxin-2 protein.

RfxCas13d was found to provide broad therapeutic benefit to TAR4/4 mice. Compared to control animals, mice infused with AAV9-CBh-RfxCas13d-mATXN2 had improved gait ($P < 0.05$; Fig. 4b), decreased kyphosis ($P < 0.05$; Fig. 4c), reduced tremors ($P < 0.05$; Fig. 4d) and increased weight ($P < 0.05$; Fig. 4e). Moreover, TAR4/4 mice treated by RfxCas13d possessed improved motor coordination and neuromuscular function, as animals infused with AAV9-CBh-RfxCas13d-mATXN2 attained higher maximum speeds on an accelerating rotarod ($P < 0.05$; Fig. 4f) and displayed improved forelimb grip strength ($P < 0.05$; Fig. 4g) relative to animals injected with the control vector. Additionally, and most strikingly, TAR4/4 mice treated by RfxCas13d were found to have a ~35-day increase in mean survival compared to the control mice (mATXN2: $61.56 \pm 13.89$ days; NTG: $26 \pm 2.6$ days; $P < 0.05$; Fig. 4h), with several treated TAR4/4 mice found to survive for over 80 days.

We next analyzed the distribution and targeting of RfxCas13d in TAR4/4 mice. Despite the progressive neurodegenerative phenotype that develops in TAR4/4 mice, RfxCas13d expression was observed in NeuN[+] cells in the MC, BS, and the SC of end-stage TAR4/4 mice (Fig. 4i and Supplementary Fig. 13). qPCR further revealed that, relative to control animals, TAR4/4 mice treated with AAV9-CBh-RfxCas13d-mATXN2 had a ~40%, a ~30% and a ~25% decrease in mATXN2 mRNA in the MC, BS, and SC, respectively ($P < 0.01$ for MC and $P < 0.05$ for the BS and SC; Fig. 4j), indicating that RfxCas13d suppressed mATXN2 mRNA expression. Critically, qPCR also revealed no significant difference in the relative abundance of hTDP-43 mRNA in the MC, BS, and SC of the treated mice versus the controls ($P > 0.05$ for all; Fig. 4k),

suggesting that RfxCas13d affected disease progression without significantly affecting the expression of hTDP-43.

Finally, we determined if RfxCas13d affected several of the neuropathological hallmarks that develop in TAR4/4 mice and are characteristic of aspects of ALS, hallmarks that include: (i) the loss of cortical and spinal cord neurons, (ii) the phosphorylation and aggregation of TDP-43, (iii) the accumulation of ubiquitinated inclusions, and (iv) the development of prominent microgliosis and astrogliosis[69].

We first analyzed neuronal viability in TAR4/4 mice. Based on an immunofluorescent analysis that was conducted on tissue from end-stage animals, TAR4/4 mice injected with AAV9-CBh-RfxCas13d-mATXN2 were found to have ~23% more NeuN[+] cells in the MC and ~36% more NeuN[+] cells in the anterior horn of the lumbar spinal cord compared to mice infused with the non-targeted vector ($P < 0.05$ for both; Fig. 4l, Supplementary Fig. 14 and Supplementary Fig. 15).

We then evaluated whether RfxCas13d affected TDP-43 phosphorylation. According to an immunofluorescent analysis conducted with a phospho-TDP-43 (pTDP-43)-specific antibody that binds to the pathological species[76], we found that mice infused with AAV9-CBh-RfxCas13d-mATXN2 had a ~61% decrease in the number of TDP-43-immunoreactive inclusions per NeuN[+] cell in the MC ($P < 0.05$; Fig. 4l, m and Supplementary Fig. 14). A subsequent western blot analysis conducted with the same pTDP-43-specific antibody further revealed that mice treated by RfxCas13d also had a ~33% decrease in pTDP-43 protein in the MC compared to the controls ($P < 0.05$; Fig. 4n and Supplementary Fig. 16).

To determine if RfxCas13d affected the deposition of ubiquitin[+] inclusions in TAR4/4 mice, we used an immunofluorescent analysis to quantify the number of ubiquitin-immunoreactive inclusions per NeuN[+] cell in the MC, finding that TAR4/4 mice treated by RfxCas13d had a ~27% decrease in the average number of these inclusions per cell compared to control animals ($P < 0.05$; Fig. 4o, p and Supplementary Fig. 14).

Last, we examined neuroinflammation, measuring that TAR4/4 mice injected with AAV9-CBh-RfxCas13d-mATXN2 had a ~36% and a ~34% reduction in the number of Iba1[+] and GFAP[+] cells within layer V of the MC, respectively ($P < 0.05$ for both; Supplementary Fig. 17), indicating that RfxCas13d also reduced the proliferation of microglia and astrocytes.

Altogether, these results demonstrate that RfxCas13d can be delivered to the nervous system by AAV9, that RfxCas13d can lower mATXN2 mRNA in vivo, and that its action in TAR4/4 mice can improve impairments, slow disease progression, and affect hallmarks of TDP-43 pathology.

## High-fidelity Cas13 and Cas7−11 can target ataxin-2

Upon binding to a target RNA, RfxCas13d – like other Cas13 effectors – gains the ability to collaterally *trans*-cleave non-target RNAs[52,54–56], an outcome that can lead to the non-specific degradation of bystander RNAs in cells. Because of this, high-fidelity (HiFi) forms of the protein have been developed that possess a reduced capacity to *trans*-cleave RNAs, but which still retain the ability to cleave their target transcript[63].

Additionally, and in contrast to the Cas13 family of enzymes, DiCas7−11, a sub-type III-E effector protein from *Desulfonema ishimotonii*, has been found as capable of mediating RNA knockdown through a crRNA-dependent mechanism that is currently believed to pose minimal risk for inducing collateral effects in cells[46]. Given their features, we sought to determine if HiFi versions of the RfxCas13d protein, as well as DiCas7−11, could be used to target ataxin-2 and if their implementation would lead to fewer non-specific effects than with the native RfxCas13d protein.

The extent to which Cas13 enzymes can collaterally cleave non-target RNAs can correlate with the expression threshold of the target transcript[52–55]. Knowing this, we evaluated the targeting capabilities of three HiFi RfxCas13d variants, RfxCas13d-N1V7, -N2V7, and -N2V8,

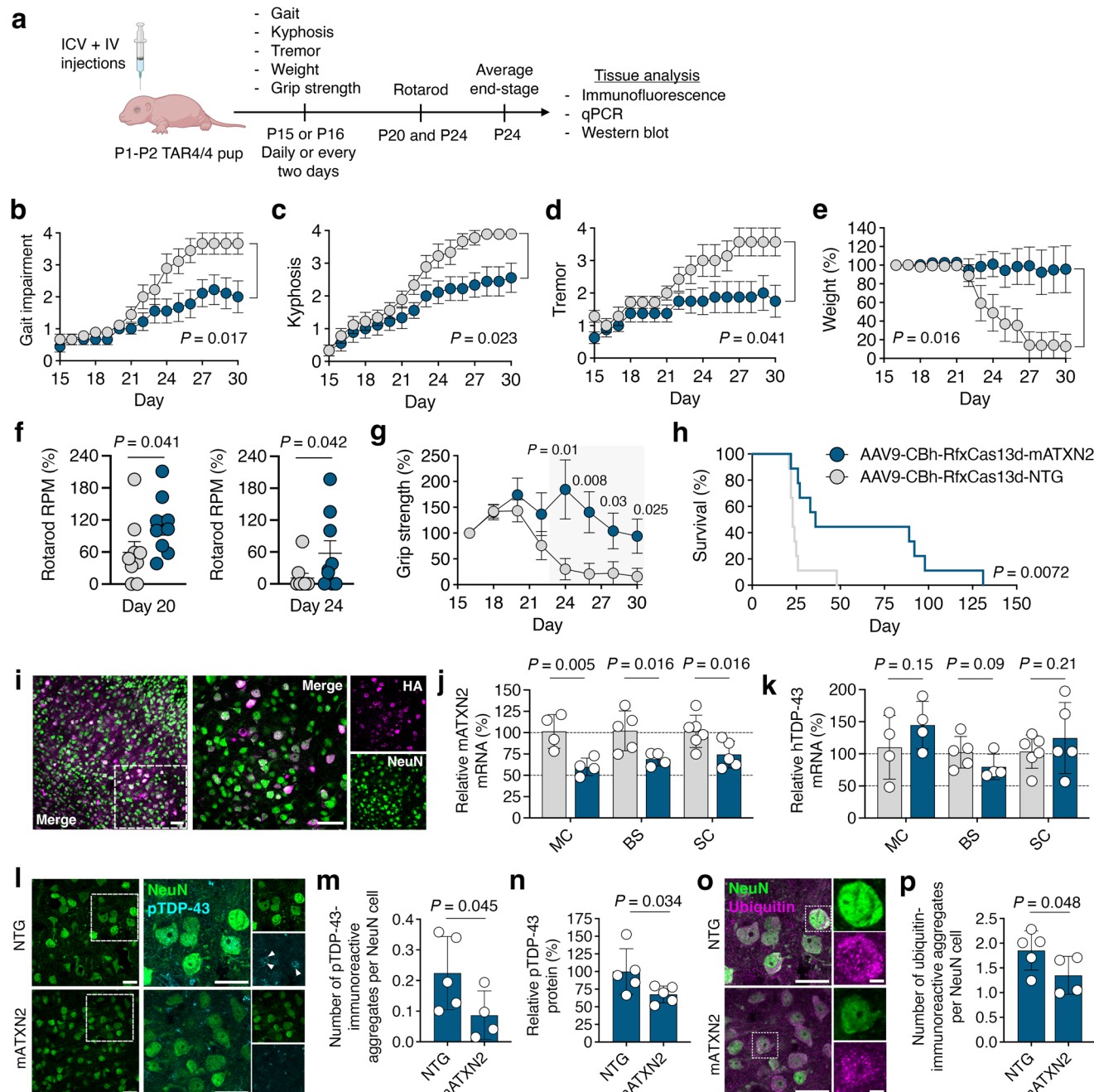

**Fig. 4 | RfxCas13d improved deficits and neuropathological hallmarks in TAR4/4 mice. a** Timeline. ICV, intracerebroventricular; IV, intravenous, **b** gait, **c** kyphosis, **d** tremor, **e** weight, **f** rotarod, **g** grip strength and **h** survival of TAR4/4 mice injected with AAV9-CBh-RfxCas13d-mATXN2 or AAV9-CBh-RfxCas13d-NTG.
**b, c, e, f, g, h** n = 9 for mATXN2, n = 9 for NTG; **d** n = 8 for mATXN2, n = 7 for NTG. Mean values for each mouse were normalized to (**b, c, d**) day 15 or (**e, f, g**) day 16 values for the same subject. **b–g** Values represent means and error bars indicate SEM. **b, c, e** two-way ANOVA; **d** mixed-effects model using a Šidák correction; **f, g** one-tailed unpaired *t*-test; **h** Log-rank test. Exact *P*-values shown.
**i** Immunofluorescence staining of the motor cortex (MC) in TAR4/4 mice. Scale bar, 50 μm. **j, k** Relative **j** mATXN2 and **k** hTDP-43 mRNA from TAR4/4 mice injected with AAV9-CBh-RfxCas13d-mATXN2. Data are normalized to TAR4/4 mice injected with AAV9-CBh-RfxCas13d-NTG (MC, n = 4; BS, n = 5 for NTG, n = 4 for mATXN2; SC, n = 6 for NTG, n = 5 for mATXN2). **l** Immunofluorescence staining of NeuN and

pTDP-43 in the MC of TAR4/4 mice with (**m**) quantification. Arrowheads indicate pTDP-43-immunoreactive inclusions. Images were captured using identical exposure conditions. Scale bar, 25 μm (n = 5 for NTG, n = 4 for mATXN2). **n** Relative pTDP-43 protein in the MC from TAR4/4 mice injected with AAV9-CBh-RfxCas13d-mATXN2. Data are normalized to TAR4/4 mice injected with AAV9-CBh-RfxCas13d-NTG (n = 5). **o** Immunofluorescence staining of NeuN and ubiquitin in the MC of TAR4/4 mice with (**p**) quantification. Images were captured using identical exposure conditions. Scale bar, (left) 25 μm and (right) 5 μm. (n = 5 for NTG, n = 4 for mATXN2). **l, m, o, p** >150 cells analyzed per biological replicate. **j, k, m, n, p** All analysis conducted on end-stage tissue from TAR4/4 mice injected on P1-P2. Values represent means and error bars indicate SD. Data were compared using a one-tailed unpaired *t*-test. Exact *P*-values shown. All data points are biologically independent samples.

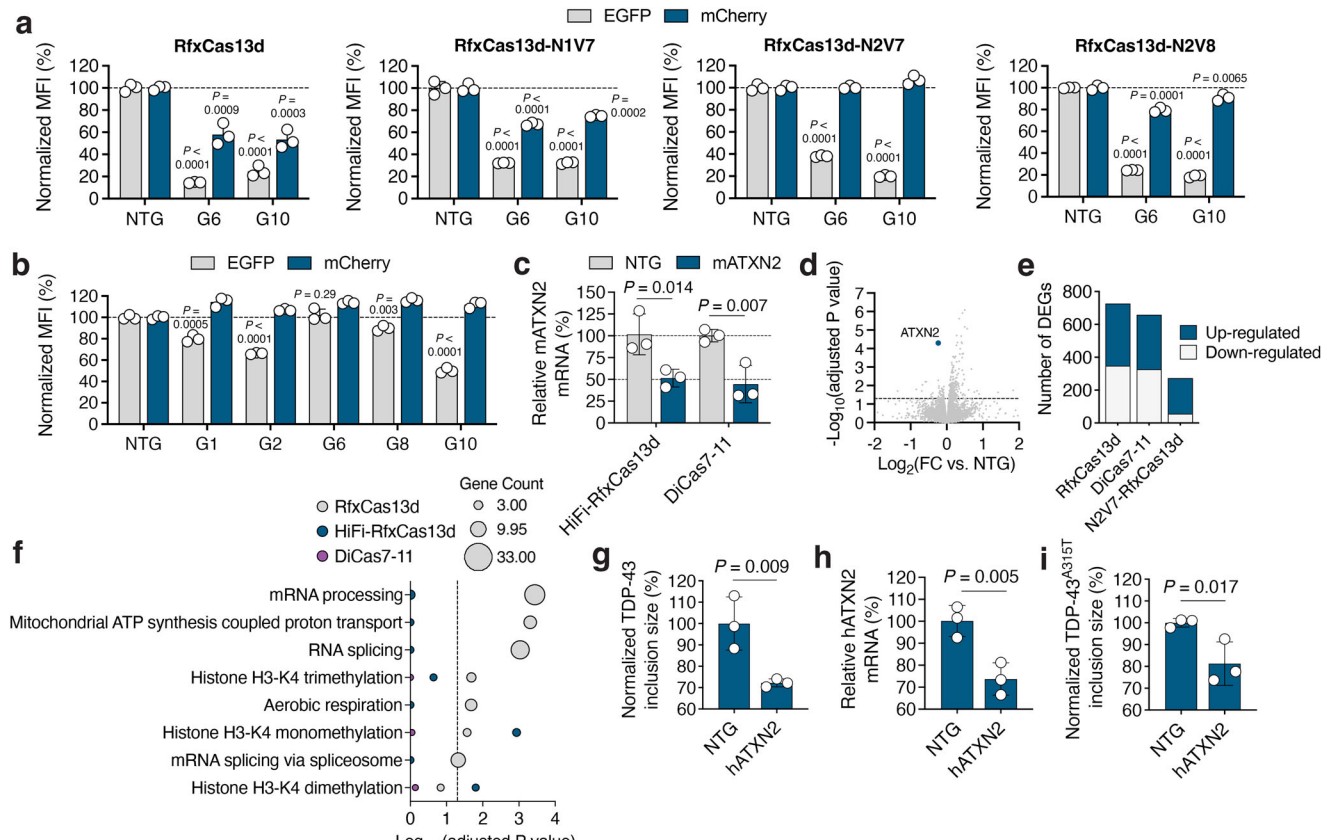

**Fig. 5 | High-fidelity RfxCas13d and DiCas7–11 can target ATXN2. a, b** EGFP and mCherry mean fluorescence intensity (MFI) in HEK293T cells transfected with mATXN2-T2A-EGFP and pCAG-mCherry with **a** an RfxCas13d variant or **b** DiCas7–11 and mATXN2-targeting crRNA. Data are normalized to MFI from cells transfected with mATXN2-T2A-EGFP and pCAG-mCherry with RfxCas13d or DiCas7–11 and a non-targeted (NTG) crRNA ($n = 3$). **c** Relative mATXN2 mRNA from Neuro2A cells transfected with RfxCas13d-N2V7 (HiFi-RfxCas13d) or DiCas7–11 and crRNA-10. Data are normalized to mATXN2 mRNA from cells transfected with each variant and a NTG crRNA ($n = 3$). **a–c** Data were compared using a one-tailed unpaired $t$-test. Exact $P$-values shown. **d** Volcano plot of RNA-seq comparing Neuro2A cells transfected with HiFi-RfxCas13d and crRNA-10 to HiFi-RfxCas13d and a NTG crRNA ($n = 3$). Line denotes FDR-adjusted $P < 0.05$. **e** Number of differentially expressed genes (DEGs) [>1-fold change (FC), FDR-adjusted $P$-value < 0.05] for each variant with crRNA-10 relative to Neuro2A cells transfected with each variant and a NTG crRNA ($n = 3$). **d, e** FDR-adjusted $P$-value was determined by a Global FDR correction

across pairwise comparisons. **f** Gene ontology (GO) and biological process (BP) term analysis. Line denotes FDR-adjusted $P < 0.05$, which was used to determine enriched terms. **g** Immunofluorescent quantification of TDP-43 inclusion size in HEK293T cells transfected with HiFi-RfxCas13d and crRNA-6 and exposed to sorbitol. Data are normalized to cells transfected with RfxCas13d and a NTG crRNA ($n = 3$). >134 cells analyzed per replicate. **h** Relative hATXN2 mRNA from HEK293T cells transfected with HiFi-RfxCas13d and crRNA-10. Data are normalized to relative hATXN2 mRNA from cells transfected with HiFi-RfxCas13d and NTG crRNA ($n = 3$). **i** Immunofluorescent quantification of TDP-43$^{A315T}$ inclusion size in HEK293T cells transfected with YFP-TDP-43$^{A315T}$ and HiFi-RfxCas13d with crRNA-6. Data are normalized to cells transfected with YFP-TDP-43$^{A315T}$ and HiFi-RfxCas13d with a NTG crRNA ($n = 3$). >81 cells analyzed per replicate. **g–i** Data were compared using a one-tailed unpaired $t$-test. Exact $P$-values are shown. Values represent means and error bars indicate SD. All data points are biologically independent samples.

in addition to DiCas7–11, in HEK293T cells, which we transiently transfected with mATXN2-T2A-EGFP alongside mCherry, whose fluorescence was used to measure collateral effects.

Consistent with studies demonstrating that targeting transcripts with high expression thresholds can lead to robust collateral effects[52–56], we measured that the native RfxCas13d protein reduced mCherry fluorescence by ~40–60% when programmed to target mATXN2-T2A-EGFP ($P < 0.0001$; Fig. 5a, Supplementary Fig. 2d and Supplementary Fig. 18). However, when paired with the same crRNAs, we observed that the HiFi forms of RfxCas13d reduced the severity of these collateral effects (Fig. 5a and Supplementary Fig. 2d), with RfxCas13d-N2V7 found as the variant most effective at decreasing mATXN2-T2A-EGFP without affecting mCherry fluorescence ($P < 0.001$ for EGFP; Fig. 5a).

Similarly, when programmed to target the same sequences as the RfxCas13d variants, we found that DiCas7–11 also decreased EGFP without diminishing mCherry fluorescence (Fig. 5b); however, for the pool of crRNAs that were examined, DiCas7–11 was observed as generally less efficient than the HiFi forms of RfxCas13d (Fig. 5b).

We next determined the ability of RfxCas13d-N2V7, hereafter referred to as HiFi-RfxCas13d, and DiCas7–11 to target the endogenous mATXN2 transcript. Neuro2A cells were thus transfected with each variant alongside crRNA-10, the mATXN2-targeting crRNA that we deployed in vivo. According to qPCR, which was conducted at 48 hr post-transfection, we found that HiFi-RfxCas13d and DiCas7–11 both lowered mATXN2 mRNA by ~50–55% ($P < 0.05$ for HiFi-RfxCas13d; $P < 0.01$ for DiCas7–11; Fig. 5c), which was on par with our measurements for the native RfxCas13d protein (Fig. 1d).

To compare their transcriptome-wide specificities, we conducted an RNA-seq analysis on Neuro2A cells transfected with RfxCas13d, HiFi-RfxCas13d and DiCas7–11 alongside their respective mATXN2-targeting crRNA. Each variant was observed to exhibit favorable specificity when programmed to target the endogenous mATXN2 transcript; however, HiFi-RfxCas13d was found to affect fewer genes (273) compared to DiCas7–11 (659 genes) and RfxCas13d (727 genes; >1-fold change [FC], FDR-adjusted $P$-value < 0.05 for all; (Fig. 5d, e and Supplementary Data 1), indicating that, within this context, HiFi-RfxCas13d possessed the highest transcriptome-wide specificity.

Using an over-representation analysis of gene ontology (GO) and biological process (BP) terms, we next compared the themes enriched for the differentially expressed genes (DEGs) by RfxCas13d, HiFi-RfxCas13d and DiCas7–11. For the native RfxCas13d protein, we observed enrichment for seven biological functions ($P < 0.05$; Fig. 5f and Supplementary Data 2), including mRNA processing, mitochondrial ATP synthesis, RNA splicing and mRNA splicing via the spliceosome, each of which has been previously linked to ataxin-2[77–79]. In addition to these functions, we also observed enrichment for histone H3-K4 monomethylation and histone H3-K4 dimethylation (Supplementary Data 2), indicating the possibility that depleting ataxin-2 with RfxCas13d affected transcription and/or its regulation. For HiFi-RfxCas13d, enrichment was observed for only two functions, histone H3-K4 monomethylation and histone H3-K4 dimethylation ($P < 0.05$; Fig. 5f and Supplementary Data 2). No significant enrichment ($P > 0.05$; Fig. 5f) was observed for any themes for DiCas7–11.

Interestingly, we found that ~52%, ~79%, and ~50% of the DEGs for RfxCas13d, HiFi-RfxCas13d, and DiCas7–11, respectively, were up-regulated (Fig. 5e). Mitochondrial ATP synthesis, mRNA processing, and RNA splicing were among the biological functions enriched within the up-regulated DEG population for RfxCas13d which, as noted above, were previously linked to ataxin-2[77–79] (Supplementary Fig. 19a and Supplementary Data 3). Histone H3-K4 monomethylation and histone H3-K4 dimethylation were the only up-regulated functions enriched for HiFi-RfxCas13d ($P < 0.05$; Supplementary Fig. 19a and Supplementary Data 3). Among the down-regulated genes, no significant themes were identified for any variant (Supplementary Fig. 19b and Supplementary Data 4).

Last, we analyzed the shared DEGs between the native RfxCas13d protein, HiFi-RfxCas13d, and DiCas7–11. In total, 67 DEGs were shared among the three variants, while 110 and 95 DEGs were shared between RfxCas13d and HiFi-RfxCas13d, and HiFi-RfxCas13d and DiCas7–11, respectively (Supplementary Fig. 20 and Supplementary Data 5). Though no significant functions ($P > 0.05$) were enriched for any shared DEGs, gene(s) related to cell adhesion, methylation, chromatin organization, and transcription were observed as among the most commonly shared for the variants (Supplementary Data 6).

Finally, because of its favorable targeting characteristics, we sought to determine if HiFi-RfxCas13d could inhibit TDP-43 aggregation when programmed to target the endogenous hATXN2 transcript. We therefore transfected HEK293T cells with HiFi-RfxCas13d and crRNA-6, the hATXN2-targeting crRNA, before exposing the cells to sorbitol for 1 hr to induce the formation of SGs. Through an immunofluorescent analysis, we measured that HiFi-RfxCas13d decreased the size of TDP-43⁺ inclusions in HEK293T cells, though to a lesser extent than previously observed for the native RfxCas13d protein, as HiFi-RfxCas13d decreased inclusion size by ~28% compared to the control cells ($P < 0.01$; Fig. 5g and Supplementary Fig. 21a). Interestingly, despite targeting the mATXN2 mRNA at a frequency equivalent to the native protein in Neuro2A cells, we measured by qPCR that the HiFi-RfxCas13d variant was less effective in HEK293T cells, decreasing the hATXN2 mRNA by ~25% at 48 hr post-transfection ($P < 0.01$; Fig. 5h).

To evaluate if HiFi-RfxCas13d could also prevent the formation of inclusions consisting of ALS-linked TDP-43 mutants, we transfected HEK293T cells with HiFi-RfxCas13d and crRNA-6 alongside a reporter expressing TDP-43^A315T linked to YFP, which enabled us to monitor the formation of pathological aggregates. Consistent with the effect observed for the inclusions formed by sorbitol, we found that HiFi-RfxCas13d reduced the average size of inclusions for TDP-43^A315T by ~20% ($P < 0.05$; Fig. 5i and Supplementary Fig. 21b). These results indicate that high-fidelity forms of Cas13 can influence a TDP-43 pathology in cell culture models.

In conclusion, we find that RNA-targeting CRISPR effector proteins can be programmed to target ataxin-2 and that lowering ataxin-2 with these platforms can mitigate aspects of TDP-43 pathology.

## Discussion

Because of their intrinsic RNase activity[39–45], their high programmability, and their ability to be used in mammalian cells to knock down the expression of a target gene[39,44,49–51,80], RNA-targeting CRISPR effector proteins are a potentially powerful class of agents for gene silencing. Here we demonstrate that two such RNA-targeting effector proteins – RfxCas13d[44,63] and DiCas7–11[46] – can be programmed to target ataxin-2, a protein whose downregulation can attenuate TDP-43-associated toxicity[20]. More specifically, we show that RfxCas13d and a high-fidelity version of it can be used to inhibit the formation of TDP-43⁺ aggregates in cell culture models of the pathology and that the in vivo delivery of an ataxin-2-targeting RfxCas13d-based system to a mouse model of a TDP-43 proteinopathy can provide broad therapeutic benefit. In particular, we observed that RfxCas13d could not only improve gait, kyphosis, tremors, weight loss, grip strength, and rotarod performance, but it could also extend lifespan and decrease the severity of several neuropathological hallmarks, including neuronal survival, the accumulation of phosphorylated TDP-43, the deposition of ubiquitinated inclusions, and microgliosis and astrogliosis. Our results thus illustrate the potential of CRISPR technology for TDP-43 proteinopathies, which can include ALS and FTD.

Though they affect different regions of the central nervous system, ALS and FTD are two conditions that are thought to exist on a continuous disease spectrum[1,6]. ALS, in particular, is a rapidly progressive disorder that affects motor neurons in the spinal cord and brain, resulting in the loss of muscle control, paralysis, and ultimately death. At present, there is no cure for ALS and current therapies provide limited benefit. While strategies for silencing dominantly inherited mutant genes causative for ALS, such as superoxide dismutase 1[49,81–83] or C9ORF72[84–86], hold promise for those familial forms of the disease, ~90% of ALS cases are sporadic, with few clear factors for targeting. This notwithstanding, ~97% of ALS cases have been characterized by inclusions consisting of ubiquitinated and hyperphosphorylated TDP-43[1–4], indicating it may play a central role in the pathophysiology of the disease, a finding that has helped to elevate its status as a therapeutic target. However, the TDP-43 protein may not be an ideal choice for silencing, as it plays a critical role in many cellular processes[15]. For this reason, attention has turned instead to the identification of modifier(s) whose targeting can attenuate the apparent toxic effects induced by aberrant TDP-43 but without affecting its underlying levels[20–24]. Ataxin-2 has emerged as one such candidate. While the exact mechanism by which ataxin-2 modifies TDP-43-associated toxicity has not been fully unraveled, strategies for targeting ataxin-2 nonetheless appear to hold potential for TDP-43 proteinopathies[26]. Notably, an antisense oligonucleotide (ASO) directed against ataxin-2 was found to lower the toxicity of TDP-43 in a transgenic mouse model of the pathology[26], with a related approach now under evaluation in a clinical trial for ALS (ClinicalTrials.gov Identifier: NCT04494256). However, despite their successes, ASOs can possess drawbacks. Among them is their transient life cycle, which can result in periods of diminished efficacy that necessitate redosing, a caveat that can impose a physical burden on patients. For this reason, we explored the utility of RNA-targeting CRISPR proteins to target ataxin-2, as these platforms are genetically encodable and can be expressed from a viral vector to continuously engage with a target mRNA, which offers a theoretical advantage over modalities with transient lifecycles that require redosing.

Cas13 proteins, however, possess a limitation that can impact their potential therapeutic implementation: after engaging with a target transcript, they undergo a conformational change that unlocks an ability to indiscriminately trans-cleave non-target RNAs[52,54–56]. This effect, known as collateral cleavage, has been observed against various endogenous targets[52,54–56] and its severity is now believed to correlate with the expression threshold of the target transcript[52,54,55]. Given its importance, we measured the collateral trans-cleavage activity of the

native RfxCas13d protein in several contexts. Notably, when programmed to target endogenous ATXN2 mRNAs in Neuro2A or HEK293T cells, we observed no reduction in the fluorescence of a surrogate mCherry reporter, which we used as a proxy for collateral cleavage; however, when tasked to target a strongly overexpressed mRNA transcript, we measured collateral cleavage, mirroring reports from the literature[52,54,55]. Such findings, in fact, have been the impetus for the development of not only high-fidelity forms of RfxCas13d and other Cas13 variants[63] but also the discovery and implementation of alternate RNA-targeting CRISPR proteins, such as Cas7−11, which is currently thought to cleave RNA by a mechanism that does not lead to collateral effects[46]. Given their potential, we compared the targeting capabilities of several HiFi variants, finding that, while HiFi-RfxCas13d could reduce collateral targeting in the context of the transient, overexpression-based reporter assay, their targeting efficiency and *trans*-cleavage capabilities fluctuated based on the HiFi variant used and the crRNA sequence employed, indicating there may exist an interplay between the stability of the crRNA, and/or its targeting sequence, and the HiFi-RfxCas13d variant itself, a relationship that requires further study.

Related to this point, while we observed that HiFi-RfxCas13d could target the mATXN2 mRNA at a frequency equivalent to the native RfxCas13d protein in Neuro2A cells, the HiFi form of the enzyme was less effective in HEK293T cells, where we observed it had a decreased ability to target hATXN2. We hypothesize this could be due to a potential incompatibility between crRNA-6, the hATXN2-targeting crRNA identified in our study, and RfxCas13d-N2V7, the HiFi-RfxCas13d variant used for these measurements. Because our initial crRNA screen was performed using the native RfxCas13d protein, we expect that future crRNA discovery efforts conducted in the context of a HiFi variant would address this concern, as subtle differences in the structure of the RfxCas13d-N2V7 protein relative to its native form could have affected its ability to effectively engage with the hATXN2 mRNA when paired with crRNA-6. To this end, in addition to utilizing computational tools capable of predicting active crRNAs for RfxCas13d and/or other Cas13 effectors[87,88], directly measuring the knockdown of the endogenous ATXN2 mRNA could prove more effective for identifying an optimal crRNA than the fluorescence-based reporter strategy used here, which we employed due to its more high-throughput nature and our previous success using it to uncover crRNAs for RfxCas13d that functioned effectively in vivo[49].

A polyQ expansion of 30−33 repeats in the ATXN2 gene is recognized as a risk factor for ALS[20,30,31]. Though the mechanism(s) underlying its relationship with TDP-43 and ALS remain incompletely understood, it has been hypothesized that the polyQ expansion might make TDP-43 more suspectable to mislocalization under stress[20]. Notably, the mouse and human models used in this study do not carry an intermediate-length repeat in their respective ATXN2 genes; thus, we did not determine whether RfxCas13d could suppress TDP-43 aggregation in the context of the polyQ expansion, an important point that will require further study, though we note that the ataxin-2-targeting ASO under evaluation in human trials is being tested in ALS patients both with and without an intermediate-length polyQ expansion (ClinicalTrials.gov Identifier: NCT04494256).

Notably, when examining whether targeting hATXN2 could influence the aggregation of ALS-linked TDP-43 protein variants, we observed that inclusions for the TDP-43 variant TDP-43$^{A315T}$ were more effectively reduced than those for TDP-43$^{G298S}$ or TDP-43$^{M337V}$. Interestingly, our analysis revealed that inclusions for TDP-43$^{G298S}$ and TDP-43$^{M337V}$ were ~2.5-fold larger in size than for TDP-43$^{A315T}$, suggesting the possibility that there could exist a relationship between the size of an inclusion and its capacity to be reduced from the targeting of ataxin-2. Future studies, however, will be needed to conclusively determine this.

The ataxin-2 protein is believed to play a prominent role in RNA metabolism. Thus, it is important to establish the tolerability of

approaches that aim to lower its expression for therapeutic purposes. For this reason, we conducted an immunohistochemical analysis to determine whether targeting mATXN2 by RfxCas13d led to adverse effects in B6SJLF1/J mice. In both the MC and the BS, two areas that saw a ~70% and a ~40% decrease in mATXN2 mRNA, respectively, we measured no difference in neuronal viability and no change in the number of cells positive for GFAP and Iba1, two neuroinflammatory markers, in mice injected with AAV9-CBh-RfxCas13d-mATXN2 versus the control. Nonetheless, a more comprehensive and longer-term study will be needed to determine the tolerability of this approach and to answer whether there exists a threshold for safely lowering ataxin-2 and whether some region(s) in the brain and/or spinal cord might be more susceptible to potential adverse effects from its lowering. To this point, it is worth noting that ATXN2 knockout mice have been reported to show no major histological abnormalities in the brain[89], though a proteomic analysis of such mice has revealed they can possess alterations in pathways related to the metabolism of branched-chain amino acids, fatty acids, and the citric acid cycle[90].

Notably, we found that the native RfxCas13d protein could broadly improve numerous therapeutic outcomes in TAR4/4 mice. Given the role that transcript threshold is thought to have on the *trans*-cleavage activity of RfxCas13d, we hypothesize that the expression level of the endogenous mATXN2 transcript in the targeted cells may have been too low to trigger adverse effects, a hypothesis supported by our RNA-seq analysis, which revealed that RfxCas13d affected the expression of 727 genes (>1-FC, FDR-adjusted *P*-value < 0.05), a number that was similar to that observed with DiCas7−11 (659), an enzyme that reportedly lacks the ability to *trans*-cleave non-target RNAs. Given the limited number of crRNAs used in this study, a more detailed investigation is needed in order to effectively compare RfxCas13d and its HiFi equivalents with DiCas7−11.

Through an over-representation analysis of GO and BP terms, seven biological functions were found to be affected in cells transfected with RfxCas13d and the mATXN2-targeting crRNA, including mRNA processing, mitochondrial ATP synthesis, RNA splicing and mRNA splicing via the spliceosome, each of which has been previously linked to ataxin-2[77−79]. We also found that the native and HiFi form of RfxCas13d affected gene(s) related to H3-K4 monomethylation and histone H3-K4 dimethylation, indicating that targeting ataxin-2 with RfxCas13d may have influenced transcription and/or its regulation, though this observation requires further study. Interestingly, no themes were significantly enriched in cells transfected with DiCas7−11, though we did observe that the RfxCas13d protein, HiFi-RfxCas13d, and DiCas7−11 shared several dozen DEGs, many of which were related to cell adhesion, methylation, chromatin organization, and transcription, though no significant terms emerged from our analysis.

To ensure widespread delivery and efficient targeting in our proof-of-concept study, we administered RfxCas13d-encoding AAV9 vector to neonatal mice via both IV and ICV injections, as combining these routes has been found to be an effective strategy for enhancing delivery to the brain and spinal cord of neonatal mice[72−74]. In particular, we injected $7 \times 10^{11}$ vg per pup and $8.4 \times 10^{10}$ vg per pup via the facial vein and the lateral ventricles, respectively. We note these are high doses. Given emerging concerns related to the potential toxicity of high-dose AAV gene therapy[91−93], it will be critical for future studies to identify the optimal dose, capsid and delivery strategy to safely and effectively target ATXN2. Related to this point, another potential challenge facing the clinical implementation of AAV vectors is their immunogenicity[94,95]. In particular, it has been well documented that immune-related responses against the AAV capsid can lower its efficacy and/or induce potential adverse events[96], a risk that can be intensified by the systemic delivery of a high-dose formulation of an AAV vector[97]. To this end, the development of strategies to control and/or mitigate these effects has emerged as an important area of investigation[95,96]. While this proof-of-concept demonstrates the

potential for RNA-targeting CRISPR technology to affect TDP-43 proteinopathy, additional work will be needed to answer several important questions beyond those already discussed, including: (i) what is the effectiveness of this approach in other models of ALS-FTD and (ii) can targeting ataxin-2 at a later stage of disease affect or reverse its progression? Additional work is also needed to identify optimal RNA-targeting CRISPR effector platforms for targeting ataxin-2. In addition to the gene silencing modalities discussed here, traditional CRISPR nucleases and next-generation DNA editing technologies[98] could also conceivably be used to target ataxin-2.

In conclusion, we show here that RNA-targeting CRISPR effector proteins can be programmed to target ataxin-2, a potent modifier of TDP-43-associated toxicity, and that the in vivo delivery of an ataxin-2-targeting Cas13 system to a mouse model of TDP-43 proteinopathy could provide broad therapeutic benefit. Our results demonstrate the potential of CRISPR technology for TDP-43 proteinopathies.

## Methods

### Ethical statement
All animal procedures were approved by the Institutional Animal Care and Use Committee (IACUC) at the University of Illinois and conducted in accordance with the National Institutes of Health Guide for the Care and Use of Laboratory Animals.

### Plasmid construction
RfxCas13d and its crRNA were expressed from the previously described[49] plasmid pAAV-CAG-RfxCas13d-U6-crRNA.

To clone the reporter plasmid mATXN2-T2A-EGFP, PCR was used to amplify the T2A-EGFP sequence from pXR001 (Addgene #109049; a gift from Patrick Hsu[44]) using the primers T2A-GFP Gibson Forward and Reverse (Supplementary Table 1). The ensuing amplicon was ligated into the EcoRV and PmeI restriction sites of pCMV-mATXN2-Myc-PolyA (MR218010, OriGene) by Gibson assembly using the Gibson Assembly Master Mix (New England Biolabs, NEB).

The plasmid encoding hTDP-43 fused to YFP (pcDNA3.2-hTDP-43-YFP) was a gift from Aaron Gitler (Addgene #84911). To construct the hTDP-43 mutant-carrying plasmids pcDNA3.2-hTDP-43$^{A315T}$-YFP, pcDNA3.2-hTDP-43$^{M337V}$-YFP, and pcDNA3.2-hTDP-43$^{G298S}$-YFP, site-directed mutagenesis was conducted using the Q5 Site-Directed Mutagenesis Kit (NEB). Briefly, primers with single nucleotide substitutions encoding the A315T (GCG to ACG), G298S (GGT to AGT), and M337V (ATG to GTG) mutations were incubated with pcDNA3.2-TDP-43-YFP and Q5 Hot Start High-Fidelity DNA Polymerase, according to the manufacturer's instructions. The ensuing reaction product was incubated with the KLD enzyme mix for 5 min at room temperature (RT) and transformed to NEB 5-alpha *Escherichia coli*.

The plasmid used for AAV vector manufacturing, pAAV-CBh-RfxCas13d-U6-crRNA, was created by VectorBuilder by replacing the CAG promoter from pAAV-CAG-RfxCas13d-U6-crRNA[49] with the chicken β-actin promoter (CBh) promoter[99,100].

To construct the plasmids encoding the HiFi-RfxCas13d (pAAV-CAG-RfxCas13d-N1V7-U6-crRNA, pAAV-CAG-RfxCas13d-N2V7-U6-crRNA, and pAAV-CAG-RfxCas13d-N2V8-U6-crRNA), PCR was used to introduce HiFi mutations into the RfxCas13d gene sequence. Each variant was amplified as two fragments from pAAV-CAG-RfxCas13d-U6-crRNA using the primers: (1) Fusion_NcoI_Forward_v2 with Fusion_N1V7_R, Fusion_N2V7_R, or Fusion_N2V8_R; and (2) Fusion_N1V7_F, Fusion_N2V7_F or Fusion_N2V8_F with Fusion_SacI_R-reverse (Supplementary Table 1). The fragments were then fused together by overlap PCR. Once assembled, each variant was ligated into the NcoI and SacI restriction sites of pAAV-CAG-RfxCas13d-U6-crRNA.

To clone the pAAV-CAG-DiCas7-11-U6-crRNA plasmid, PCR was used to amplify the U6-crRNA scaffold and the DiCas7–11 gene sequence from pDF0114 (Addgene #172508) and pDF0159 (Addgene #172507), respectively. Both plasmids were gifts from Omar Abudayyeh and Jonathan Gootenberg. The primers used to amplify the gene sequences were: NheI-U6_F and KpnI-polyT7-11_R for the U6-crRNA scaffold, and F_DiCas7–11 and R_DiCas7–11 for the DiCas7–11 gene sequence. PCR was then used to separately amplify the sequence encoding the HA epitope from pAAV-CAG-RfxCas13d-U6-crRNA using the primers F_BsiWI_3HA and R_3HA. The amplicon encoding the U6-crRNA scaffold was ligated into the NheI and KpnI restriction sites of pAAV-CAG-RfxCas13d-U6-crRNA. The amplicons encoding DiCas7–11 and the HA epitope were fused by overlap PCR and ligated into the HindIII and AgeI restriction sites of pAAV-CAG-RfxCas13d-U6-crRNA, replacing RfxCas13d and its U6-crRNA sequence.

crRNA targeting sequences were cloned into their designated plasmids, as described[49]. Briefly, oligonucleotides encoding the crRNA sequences were custom synthesized (Integrated DNA Technologies) and incubated with T4 polynucleotide kinase (NEB) for 30 min at 37 °C, annealed at 95 °C for 5 min, and then cooled to 4 °C at a rate of −0.1 °C/s. Annealed oligonucleotides were then ligated into the BbsI restriction sites of pAAV-CAG-RfxCas13d-U6-crRNA, pAAV-CAG-RfxCas13-U6-crRNA and pAAV-CAG-HiFi-RfxCas13d-U6-crRNA or the BsaI restriction sites of pAAV-CAG-DiCas7-11-U6-crRNA.

Sanger sequencing (ACGT) was used to confirm the identity of all plasmids. All primer sequences are provided in Supplementary Table 1.

### Cell culture
HEK293T cells (American Type Culture Collection [ATCC], CRL-3216) and Neuro2A cells (a gift from Pablo Perez-Pinera; ATCC, CCL-131) were maintained in Dulbecco's modified Eagle's medium (DMEM; Corning) supplemented with 10% (v/v) fetal bovine serum (FBS; Thermo Fisher Scientific) and 1% (v/v) antibiotic-antimycotic (Thermo Fisher Scientific) in a humidified 5% $CO_2$ incubator at 37 °C. For all transfections, cells were seeded onto 24-well plates at an average density of $2 \times 10^5$ cells per well.

Cells were transfected with 450 ng of pAAV-CAG-RfxCas13d-U6-crRNA, pAAV-HiFi-RfxCas13d-U6-crRNA, or pAAV-DiCas7-11-U6-crRNA with or without 50 ng of mATXN2-T2A-EGFP and 50 ng of pCAGm-Cherry using Lipofectamine 3000 (ThermoFisher Scientific) according to the manufacturer's instructions.

### Flow cytometry
At 48 hr after transfection, cells were harvested, washed with phosphate-buffered saline (PBS), and strained into single-cell suspensions using Falcon Round-Bottom Polystyrene Test Tubes with Cell Strainer Snap Caps (Corning). Fluorescence (FITC and PE/Texas Red) was measured using a BD LSRFortessa Cell Analyzer (Roy J. Carver Biotechnology Center Flow Cytometry Facility, Univ. Illinois). A total of 20,000 events were recorded for each replicate, and data were analyzed using FlowJo v10 (FlowJo, LLC).

### qPCR
At 48 hr after transfection, RNA was extracted from cells using the PureLink RNA Mini Kit (Invitrogen) and converted to complementary DNA (cDNA) using the iScript cDNA Synthesis Kit (Bio-Rad) according to the manufacturers' instructions. qPCR was conducted using 20−40 ng of cDNA per reaction using iTaq Universal SYBR Green Supermix (Bio-Rad). Measurements for each biological replicate were conducted in technical triplicates using validated primers for ataxin-2[101]. Values were compared to GAPDH, and the average fold-change was calculated using the $2^{\Delta\Delta CT}$ method. All primer sequences are provided in Supplementary Table 1.

Tissues from injected animals were dissected and RNA was extracted and analyzed by qPCR as described above for ataxin-2. hTDP-43 mRNA was measured using the PrimeTime Gene Expression Master Mix (IDT) with the following TaqMan probes: human TDP-43

(Hs00606522_m1; ThermoFisher Scientific) and mouse Actb (Mm02619580_g1; ThermoFisher Scientific).

## Western blot

Cells and tissues were lysed by radioimmunoprecipitation assay (RIPA) buffer (0.2% IGEPAL CA-620, 0.02% SDS with VWR Life Science Protease Inhibitor Cocktail). For the detection of the pTDP-43 protein, the Halt Phosphatase Inhibitor Cocktail (Thermo Scientific, 78420) was added to RIPA buffer.

Protein concentration was determined using the DC Protein Assay Kit (Bio-Rad). Protein (30 µg for lysate from cells and 50 µg for lysate from tissues) was electrophoresed by SDS-PAGE and electrophoretically transferred onto a nitrocellulose membrane (0.45 µm pore size) in transfer buffer [20 mM Tris-HCl, 150 mM glycine, and 20% (v/v) methanol] for 2 hr at 100 V. To probe for ataxin-2, membranes were blocked with 2% bovine serum albumin (BSA) in PBS with 0.1% Tween 20 (PBS-T). To probe for pTDP-43, membranes were blocked with 2% BSA in tris-buffered saline [TBS; 10 mM Tris-HCl and 150 mM NaCl (pH 7.5)] with 0.1% Tween 20 (TBS-T). Membranes were blocked for 1 hr at RT and then incubated with primary antibodies in blocking solution at 4 °C overnight.

The following primary antibodies were used: rabbit anti-ataxin-2 polyclonal antibody (1:500; Proteintech, 21776-1-AP) for mouse ataxin-2; rabbit anti-ataxin-2 polyclonal antibody (1:2000; Novus Biologicals, NBP1-90063) for human ataxin-2; rabbit anti-pTDP-43 (pS409/410) polyclonal antibody (1:1000; CosmoBio, CAC-TIP-PTD-P07); and rabbit anti-β-actin monoclonal antibody (1:1000; Cell Signaling Technology, 4970 L).

Membranes were then washed three times with PBS-T or TBS-T for ataxin-2 or pTDP-43, respectively, and incubated with the secondary antibody, goat anti-rabbit horseradish peroxidase conjugate (1:4000; Thermo Fisher Scientific, 65–6120), in blocking solution for 1 hr at RT. Membranes were then washed again three times with either PBS-T or TBS-T, developed using SuperSignal West Dura Extended Duration Substrate (ThermoFisher Scientific) and visualized by automated chemiluminescence using ChemiDoc XRS+ (Bio-Rad). Band intensity was quantified using Image Lab Software (Bio-Rad) and normalized to the reference protein in each line.

After probing membranes for ataxin-2 or pTDP-43, membranes were stripped and re-probed for β-actin. Membranes were incubated in 10 mL of stripping solution [acetic acid and 0.5 M NaCl (pH to 2.5)] for 10 min at RT and subsequently treated with 10 mL of neutralizing solution (0.5 M NaOH) for 1 min. Membranes were then washed twice with their respective washing buffers and stained with primary antibodies, as described above.

## SG assays

For SGs induced by sorbitol, HEK293T cells were transfected with 1 µg of pAAV-CAG-RfxCas13d-U6-crRNA, pAAV-HiFi-RfxCas13d-U6-crRNA, or pAAV-DiCas7-11-U6-crRNA. At 24 h post-transfection, cells were transferred to poly-D lysine-coated coverslips (Electron Microscopy Sciences) where, after another 24 h, they were then incubated with 0.4 M sorbitol (Sigma-Aldrich) at 37 °C for 1 h. Cells were then fixed with chilled 100% methanol (v/v) for 10 min at RT and subsequently washed twice with PBS.

For assays involving mutant forms of hTDP-43, HEK293T cells were transfected with 800 ng of pAAV-CAG-RfxCas13d-U6-crRNA, pAAV-HiFi-RfxCas13d-U6-crRNA, or pAAV-DiCas7-11-U6-crRNA with 200 ng of pcDNA3.2-hTDP-43^A315T-YFP, pcDNA3.2-hTDP-43^M337V-YFP or pcDNA3.2-hTDP-43^G298S-YFP. At 24 h after transfection, cells were transferred to poly-D lysine-coated coverslips (Electron Microscopy Sciences) and allowed to adhere. After another 24 h, cells were then fixed with chilled 100% methanol (v/v) for 10 min at RT and then washed twice with PBS.

To image SGs, fixed cells were incubated with blocking solution [10% (v/v) donkey serum (Abcam) and 0.1% Triton X-100] for 1 h at RT and then stained with primary antibody in blocking solution for 1 h at RT. Cells were then washed three times with PBS and incubated with a secondary antibody for 1 h at RT. Following the incubation, coverslips were washed three times with PBS and mounted onto slides using VECTASHIELD HardSet Antifade Mounting Medium (Vector Laboratories).

Slides were imaged using a Leica TCS SP8 confocal microscope and a Zeiss Observer Z1 microscope (both at the Beckman Institute for Advanced Science and Technology, Univ. Illinois) and an AxioScan.Z1 (Carl R. Woese Institute for Genomic Biology, Univ. Illinois). Image analysis was performed using CellProfiler software as described below.

The following primary antibodies were used: rabbit anti-G3BPI (1:500; MLB International, RN048PW), mouse anti-TDP-43 (1:100; Abnova, H00023435-M01), and chicken anti-HA (1:400; Abcam, ab9111).

The following secondary antibodies were used: donkey anti-rabbit Cy3 (1:400; Jackson ImmunoResearch, 711-165-152), donkey anti-mouse Alexa Fluor 488 (1:200; Jackson ImmunoResearch, 715-545-150), and donkey anti-chicken Alexa Fluor 647 (1:400; Jackson ImmunoResearch, 703-605-155).

## Injections

All animal procedures were approved by the Institutional Animal Care and Use Committee (IACUC) at the University of Illinois and conducted in accordance with the National Institutes of Health Guide for the Care and Use of Laboratory Animals.

Transgenic TAR4/4 mice were generated by breeding male and female hemizygous TDP-43 mice (B6;SJL-Tg(Thy1-TARDBP)4Singh/J; Jackson Laboratory, Stock No. 012836). Homozygosity was determined by PCR using genomic DNA purified from toe clips. The PCR reaction was conducted using repliQa HiFi ToughMix (Quantabio) using primers (Supplementary Table 1) and procedures developed by the Jackson Laboratory. Background-matching mice were generated by breeding male and female B6SJLF1/J mice (B6SJLF1/J; Jackson Laboratory, Stock No. 100012).

ICV and IV injections were conducted on P1-P2 TAR4/4 mice or P1-P2 B6SJLF1/J mice, as described[72]. Briefly, for ICV injections, pups were infused via each lateral ventricle with $4.2 \times 10^{10}$ vg of AAV9-CBh-RfxCas13d-mATXN2 or AAV9-CBh-RfxCas13d-NTG in up to 2 µL of saline using a 10 µL Hamilton syringe with a 30-gauge needle. For IV injections, pups were infused via the facial vein with $7 \times 10^{11}$ vg of AAV9-CBh-RfxCas13d-mATXN2 or AAV9-CBh-RfxCas13d-NTG in up to 50 µL of saline using a 100 µL Hamilton syringe with a 33-gauge needle. Male and female mice were used indistinctly.

AAV vectors were manufactured by Vector Biolabs.

## Behavior

All measurements and scoring were performed by a blinded investigator as described[26,69]. Gait, kyphosis, and tremors were analyzed daily beginning at P15. Gait impairment was scored as follows: (0) mice show no impairment; (1) mouse has a limp while walking; (2) mouse has either a severe limp, a lowered pelvis, or point feet away from their body while walking; (3) mouse has either difficulty moving forward, shows minimal joint movement, does not use its feet to generate forward motion, has difficulty staying upright, or drags its abdomen on the ground; (4) mouse has reached end-stage. Mice were euthanized once they scored a 3 for the first time.

Kyphosis was scored as follows: (0) the mouse has the ability to straighten the spine when walking; (1) the mouse has mild curvature of the spine but has the ability to straighten it, (2) the mouse has mild curvature of the spine but does not have the ability to fully straighten it; (3) mouse has curvature of the spine that is maintained while

walking or sitting; (4) mouse has reached end-stage. Mice were euthanized once they scored a 3 for the first time.

Tremor was scored as follows: (0) not observed; (1) mild tremor while moving; (2) severe tremor while moving; (3) severe tremor at rest and while moving; (4) mouse has reached end-stage. Mice were euthanized once they scored a 3 for the first time.

Motor function was measured on P20 and P24, while forelimb grip strength was measured every two days beginning at P16. Motor function was determined using an accelerating Rotamex-5 (Columbus Instruments). Briefly, animals were placed on a gradually accelerating apparatus (2.5 RPM to 25 RPM over 115 s), and the average RPM achieved before falling was recorded for each mouse during a three-trial session. Grip strength was measured using a Grip Strength Meter (Harvard Apparatus). Mice were scruffed and put in position to firmly latch onto a pull bar with their forelimbs and then pulled in the opposite direction. The average maximum force (Newtons) exerted by each animal before their release of the bar was recorded in three trial sessions. Animals that reached end-stage received a 0 score for rotarod and grip strength.

Kaplan-Meier survival analysis was conducted using the artificial end-points for each animal, which was determined when: (1) animals reached a score of 3 for gait, tremor, or kyphosis, (2) animals were unable to right themselves within 10 sec of being placed on their back, or (3) animals lost 20% of their peak weight.

## Immunohistochemistry

The brain and spinal cord were harvested following transcardial perfusion with PBS. The harvested tissues were subsequently fixed with 4% paraformaldehyde (v/v) at 4 °C and sliced into 40-μm sagittal and axial sections using a CM3050 S cryostat (Leica).

Tissue sections were washed three times with PBS and incubated in blocking solution [PBS with 10% (v/v) donkey serum (Abcam) and 1% Triton X-100] for 2 hr at RT. Tissue sections were then stained with primary antibodies in a blocking solution for 72 hr at 4 °C. Sections were washed three times with PBS and incubated with secondary antibodies in a blocking solution for 2 hr at RT. Following the incubation with the secondary antibodies, sections were washed three times with PBS and then mounted onto slides using VECTASHIELD HardSet Antifade Mounting Medium (Vector Laboratories). Slides were imaged using a Leica TCS SP8 confocal microscope (Beckman Institute for Advanced Science and Technology, Univ. Illinois) and an AxioScan.Z1 (Carl R. Woese Institute for Genomic Biology, Univ. Illinois). Image analysis was performed using CellProfiler software and Zeiss ZEN lite software.

The following primary antibodies were used: rabbit anti-HA (1:500, Cell Signaling Technology, C29P4), chicken anti-GFAP (1:1000, Abcam, ab4647), goat anti-Iba1 (1:800, NOVUS, NB100-1028SS), mouse anti-NeuN (1:500, EMD Millipore Corp, MAB377), mouse anti-pTDP-43 (1:3000, CosmoBio, ps409/410), rabbit anti-ubiquitin (1:200, Proteintech, 10201-2-AP), and chicken anti-NeuN (1:500, EMD Millipore Corp, ABN91).

The following secondary antibodies were used: donkey anti-rabbit Cy3 (1:200, Jackson ImmunoResearch, 711-165-152), donkey anti-mouse Alexa Fluor 488 (1:125, Jackson ImmunoResearch, 715-545-150), donkey anti-chicken Alexa Fluor 647 (1:125, Jackson ImmunoResearch, 703-605-155), and donkey anti-goat Alexa Fluor 647 (1:200, Jackson ImmunoResearch, 705-605-147).

## Image analysis

Quantification of cell numbers, inclusion counts, and inclusion size was performed using CellProfiler 4.2.1 (Broad Institute, Cambridge, MA)[64]. All imaging was performed by a blinded investigator and the same pipeline was used to analyze all condition(s) and/or group(s) from the same experiment.

Parameters for cell and inclusion size, as well as the thresholding and filtering methods, were implemented as previously described[102].

Briefly, cells were first identified using the Identify Primary Objects module. For sorbitol-treated cells, individual cells were identified by TDP-43 immunostaining using a global, two-class Otsu thresholding method. For cells transfected with the hTDP-43-YFP reporter plasmid, cells were identified by HA immunostaining using the global, robust background thresholding method. The shape of the cells was then used to distinguish clumped objects and to draw dividing lines between them. The cell diameter setting was 50 to 180 pixels for sorbitol-treated cells, 50 to 150 pixels for TDP-43$^{A315T}$, and 60 to 180 pixels for TDP-43$^{M337V}$ and TDP-43$^{G298S}$.

Prior to quantifying inclusions, images were filtered to enhance foci. As previously described[103], foci were defined as bright speckles by the Identify Primary Objects module, which distinguished clumped objects and segmented them by foci intensity. For sorbitol-treated cells, inclusions were identified using the global, two-class Otsu thresholding method and a thresholding correction factor of 5. TDP-43$^+$ foci ranged from 3 to 25 pixels in diameter, while G3BPI$^+$ foci ranged from 1 to 35 pixels in diameter. The Relate Objects module was then used to colocalize G3BPI$^+$ foci and TDP-43$^+$ foci, which were in turn mapped to their respective cells to quantify the number of cells carrying foci for both G3BPI and TDP-43. This number was divided by the total number of analyzed cells to determine the percentage of cells with G3BPI$^+$ and TDP-43$^+$ foci. For sorbitol-treated cells, we analyzed >300 cells per replicate for RfxCas13d and >134 cells per replicate for HiFi-RfxCas13d.

To identify YFP$^+$ inclusions, the global, two-class Otsu thresholding method was used for TDP-43$^{A315T}$ using a diameter range of 5 to 35 pixels. For TDP-43$^{M337V}$ and TDP-43$^{G298S}$ conditions, the global, robust background thresholding method was used to identify YFP$^+$ inclusions in the range of 5 to 40 pixels. Once identified, inclusions were mapped to each cell using the Relate Objects module and the number inclusions per cell and the area occupied by inclusions were determined. At least 81 cells were quantified per replicate for cells expressing YFP$^+$ inclusions.

To quantify images from tissue sections, cells were identified using the Identify Primary Objects module with the global, two-class Otsu thresholding method for GFAP$^+$ and NeuN$^+$ immunostaining or the global, robust background thresholding method for Iba1$^+$ and HA$^+$ cells. We analyzed: >900 GFAP$^+$ cells and >600 Iba1$^+$ cells per replicate for the neuroinflammatory analysis in TAR4/4 mice; >150 NeuN$^+$ cells per replicate in MC for the viability analysis in TAR4/4 mice; and >80 GFAP$^+$ cells, >218 Iba1$^+$, and >304 NeuN$^+$ cells per replicate for the tolerability measurements in B6SJLF1/J mice. Cells identified by HA were mapped to NeuN$^+$ cells using the Relate Objects module to find the percentage of HA$^+$ neurons ( >104 NeuN$^+$ cells were analyzed per replicate to quantify transduction).

For SC NeuN$^+$ viability analysis in TAR4/4 mice, >105 cells were analyzed per replicate (both anterior horns per section with >2 sections per replicate) by a blinded investigator.

To identify inclusions in NeuN$^+$ cells in the MC in TAR4/4 mice, objects identified as neurons were used as a mask for the pTDP-43 and ubiquitin immunostaining. Prior to quantifying inclusions, images were filtered to enhance foci. Then, to find inclusions, the Identify Primary Objects module was used with the global two-class Otsu thresholding method using foci intensity to distinguish clumped objects. The number of foci per image was quantified and this number was divided by the total number of NeuN$^+$ cells identified ( >150 cells were analyzed per replicate).

## RNA sequencing

Library construction was performed by the Roy J. Carver Biotechnology Center (Univ. Illinois) as previously described[49]. Briefly, libraries were prepared with the Kapa Hyper Stranded mRNA library kit (Roche), quantitated by qPCR, pooled, and sequenced on two SP lanes for 101 cycles from one end of the fragments on a NovaSeq 6000 with V1.5 sequencing kits. FASTQ files were generated and demultiplexed

with the bcl2fastq v2.20 Conversion Software (Illumina). The quality of the demultiplexed FastQ files was evaluated using FastQC (version 0.11.8). Average per-base read quality scores were high quality (over 30 in all samples) and no adapter sequences were found.

RNA-seq analysis was conducted by the High-Performance Biological Computing Core (Univ. Illinois) as previously described[49]. Salmon3 version 1.5.2 was used to quasi-map reads to the transcriptome and to quantify the abundance of each transcript. Transcriptomes were indexed using the decoy-aware method in Salmon with the entire *Mus musculus* transcriptome file (GCF_000001635.27_GRCm39_genomic.-fna) from NCBI FTP site as the decoy sequence. Then quasi-mapping was performed to map reads to the transcriptome with additional arguments: --seqBias and --gcBias were used to correct sequence- and GC-specific biases, --numBootstraps=30 was used to compute bootstrap transcript abundance estimates and --validateMappings and --recoverOrphans were used to improve the accuracy of mappings.

Gene-level counts were estimated based on transcript-level counts using the "lengthScaledTPM" method from the tximport package to provide accurate gene-level counts estimates and to keep multi-mapped reads in the analysis. Eight groups, including the RfxCas13d, RfxCas13d-N2V7 and DiCas7–11 groups and two additional control groups, were included in the RNA-seq analysis. DEG analysis was performed using the limma-trend method using a model for all eight groups, with 10 pairwise comparisons subsequently pulled from the model. An FDR adjustment was then conducted globally across the pairwise comparisons. The processed RNA-seq data for the analysis is available in Supplementary Data 1.

Overrepresentation analysis and functional annotation was performed using DAVID[104] to identify the enriched biological terms associated with the differentially expressed genes. Biological processes from the Gene Ontology (GO) database with FDR *P*-values < 0.05 were determined as enriched terms. Venn diagrams were produced using BioVenn[105].

### Statistical analysis and reproducibility

Statistical analysis was performed using GraphPad Prism 9. For in vitro studies, mATXN2-T2A-EGFP fluorescence, mCherry fluorescence and TDP-YFP fluorescence, ATXN2 mRNA and ataxin-2 protein, as well as inclusion sizes, numbers and percentages were compared using an unpaired one-tailed *t*-test. For in vivo studies, ataxin-2 mRNA, hTDP-43 mRNA, neuronal viability, microgliosis and astrogliosis, pTDP-43 protein and NeuN⁺ cells with pTDP-43- and ubiquitin-immunoreactive inclusions were compared using an unpaired one-tailed t-test; gait impairment, kyphosis, grip strength and weight were compared by two-way analysis of variance (ANOVA); tremor was analyzed by a mixed effects model (note: a mixed effects model was used due to the unequal sizes of the groups for this specific measurement); rotarod was compared using an unpaired one-tailed *t*-test. Survival was analyzed by Kaplan-Meier analyses using the Log-rank test.

For all in vitro experiments, three independent biological replicates were used at minimum. No statistical method was used to predetermine the sample size for the in vitro experiments. For all qPCR measurements, if no or minimal amplification was observed for housekeeping transcripts, the amplification was repeated. If no or minimal application of the housekeeping transcript was observed again, the sample was omitted from the analysis. For the SG analysis, images that possessed less than two cells by Cell Profiler and/or had an artifact that posed a risk of confounding the analysis were excluded from the computational pipeline.

For in vivo experiments, the expected effect and error were informed from published literature[26]. From this information, the target sample size for each experiment was determined by power calculations using α = 0.05 and β = 0.80. The sample size reflects the number of independent biological replicates. No data were excluded from the behavior study. For quantification of the neuropathological hallmarks,

images that possessed artifacts that could be misidentified by CellProfiler as cells and/or inclusions were excluded from the computational pipeline. Only one mouse, which had a severely undersized brain relative to its littermates and was injected with AAV9-CBh-RfxCas13d-NTG was removed from the analysis. This animal was replaced by another mouse.

### Reporting summary

Further information on research design is available in the Nature Portfolio Reporting Summary linked to this article.

## Data availability

Source Data are provided as a Source Data file. The unprocessed fluorescence microscopy images for Figs. 2, 3 and 4 are available from Figshare at https://doi.org/10.6084/m9.figshare.24187503, while the unprocessed images for the figures in the Supplementary Information are available at https://doi.org/10.6084/m9.figshare.24195102. RNA-seq data have been deposited in the NCBI Gene Expression Omnibus and are accessible through GEO Series accession no. GSE244306. All Source Data is also available upon request from the corresponding author. Source data are provided with this paper.

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

## Acknowledgements

This work was supported by the NIH/NINDS (1U01NS122102-01A1 and 1R01NS123556-01A1, both to T.G.), the NIH/NIGMS (5R01GM141296 to T.G.), the Muscular Dystrophy Association (MDA602798 to T.G), the Simons Foundation (887187 to T.G.), the Parkinson's Disease Foundation (PF-IMP-1950 to T.G.) and the Judith & Jean Pape Adams Foundation (to T.G.). M.A.Z.C. was supported by the NIH/NIBIB (T32EB019944), the Mavis Future Faculty Fellows Program, and an Aspire Fellowship from the University of Illinois Urbana-Champaign. T.S. was supported by a Summer Undergraduate Research Fellowship from the School of Molecular & Cellular Biology at the University of Illinois Urbana-Champaign. Cartoons were created with BioRender.

## Author contributions

T.G. and M.A.Z.C. conceived of the study. M.A.Z.C. and S.Z. designed and cloned the plasmids. M.A.Z.C. and S.Z. conducted flow cytometry. M.A.Z.C. and H.J.M. conducted qPCR measurements. M.A.Z.C. and N.S.A performed the western blots. M.A.Z.C. conducted immunocytochemistry. M.A.Z.C., J.E.P., and H.J.M. bred and genotyped animals. M.A.Z.C. and J.E.P. performed injections. H.J.M. conducted behavior measurements as a blinded investigator. M.A.Z.C. and H.J.M. harvested and dissected tissues. M.A.Z.C., H.J.M., and T.J.S. sectioned tissue and conducted immunohistochemistry. M.A.Z.C. and T.J.S. acquired images. M.A.Z.C. and T.J.S. created pipelines for image analysis. T.G. and M.A.Z.C. wrote the manuscript with input from all authors.

## Competing interests

T.G. and J.P. filed a pending international patent application (PCT/US2022/036494) on the methods used in this manuscript. No part of this study is included in the filed patent application. The authors declare no competing interests.
