## [Peer Review File · Nature Communications]

Reviewers' Comments:

Reviewer #1:

Remarks to the Author:

A common feature of ALS and FTD is abnormal aggregation of TDP-43 protein in cytoplasmic inclusions. One way to reduce the toxicity associated with TDP-43 is to reduce the level of ataxin-2 protein, which is involved in the formation of stress granules. In the manuscript by Zeballos MA et al., the authors used RNA-targeting CRISPR-Cas systems to downregulate ataxin-2 expression in cellular models (HEK293T, Neuro2A) and in a mouse model of TDP-43 proteinopathy. They demonstrated successful knockdown of ataxin-2, resulting in a reduction in the number and size of TDP-43-positive inclusions and an impressive reduction in TDP-43 pathology in vivo.

Overall, this is an interesting study that confirms the role of ataxin-2 as a modifier of TDP-43 toxicity and its potential as a therapeutic target for TDP-43-related proteinopathies. The main novelty of this work is the demonstration that RNA-targeting CRISPR technology can be used to effectively downregulate ataxin-2 levels. What is missing here, however, is an expansion of our knowledge of the mechanism of action and a deeper look at the effects caused by ATXN2 silencing. Since the therapeutic strategy of silencing ataxin-2 (using ASO) in ALS patients is already in clinical trials, more mechanistic studies would be expected from subsequent works.

The manuscript is well written, the data are generally strong and well presented. There are several issues that need to be addressed.

Major

1. Since the intermediate-length Ataxin-2 polyQ repeat expansions are significantly associated with ALS – the question remains: what was the length of the CAG repeats in the ATXN2 locus in the models used in this study? Whether the length of the polyQ domain can affect the efficiency of TDP-43 aggregates reduction and the phenotype improvement (some level of ataxin-2 still remains in the cells, polyQ expansion of Ataxin-2 increases its half-life).
2. The efficacy of crRNA on the endogenous mATXN2 transcript in Neuro2A cells was lower than in transfected HEK293T cells. Since we have no information about the stability of the ataxin-2 protein (half-life), WB analysis of the protein level should be performed to determine its level at a given time point.
3. Similarly in HEK293T cells with induced formation of stress granules the authors observed 40% reduction of ATXN2 mRNA level after 48h - what was the level of ataxin 2 protein at that time? Please mark the time points in Fig 2a. The signals are hardly visible in the photos – please improve their quality/size. Arrows showing presence or absence of inclusions are confusing. Fig. 2c-f How many cells were counted?
4. In the experiment using three TDP-43 variants with mutations in the region responsible for protein-protein interaction - what does the result mean? (especially in the context of possible interactions with ATXN2 protein?)
5. In an in vivo experiment, the authors got spectacular results: improvement in behavioral tests, extended lifespan, reduced inflammation.... The study was done with RfxCas13d which, as they showed in Neuro2A cells, can silence other transcripts nonspecifically (over 700). Among them, about 50% were upregulated - how can this be explained? The supplementary Excel file with a list of transcripts from RNAseq is not very informative. It would be interesting to see if the most altered genes are random or clustered into some pathways? What fraction of altered transcripts are shared by RfxCas13d, HiFi-RfxCas13d and DiCas7-11?
6. Ataxin 2 is an RNA-binding protein with multiple roles in RNA metabolism and we do not know the results of ATXN2 depletion. An important question that the authors should try to answer in this work is to what level can ataxin 2 be safely silenced while achieving a reduction in TDP-43 aggregate formation and its mislocalization?
7. Discussion section: "RNA-targeting platforms offer the theoretical advantage of reversibility" – for AAV vector-based CRISPR systems that provide stable expression over a long period of time (episomal expression), reversibility is still a matter of debate (unlike ASO).

Minor

1. Page 8 (at the bottom): please correct the reference to Figure S3 instead of Figure S2
2. Please correct Fig S3: ATXN2 instead of ATAXN2; there are no white stars in the figure
3. Fig.3 I suggest adding a timeline with an overview of the study (timing of AAV injection, behavioral testing, experimental endpoints).

Reviewer #2:

Remarks to the Author:

The authors described the development and evaluation of ataxin-2 mRNA targeting CRISPR platforms (Cas13 and Cas7-11) to mitigate a TDP-43 proteinopathy. Preliminary screening and experiments were performed to identify the crRNAs knocking down ataxin-2 efficiently. The authors further demonstrated that knocking down ataxin-2 mRNA with selected crRNA and RfxCas13d attenuated the formation of TDP-43 aggregates in sorbitol-treated HEK293T cells. The therapeutic benefit of targeting ataxin-2 by RfxCas13d in a transgenic mouse model of a TDP-43 proteinopathy and specificity improvement in vitro using high-fidelity Cas13 and Cas7-11 showed the potential of RNA-targeting CRISPR technology to mitigate TDP-43 proteinopathy.

Overall, this is an exciting proof-of-concept study demonstrating the promise and significance of ataxin-2 mRNA targeting CRISPR platforms delivered by viral vector for treating TDP-43 proteinopathies, which is a complementary strategy to ASO-mediated ataxin-2 knockdown and shows some advantages on long-lasting benefits.

However, several aspects need to be studied to give clearer conclusions and interpretations. It would also be helpful to address or discuss translation considerations or possible hurdles to the readers.

Specific comments:

1. On page 5 (the first set of results) and in figure 1, crRNAs targeting mouse ataxin-2 mRNA were screened based on EGFP protein intensity which expressed/fused to mATXN2 by T2A peptide. Why were the ten crRNA candidates chosen within exon 5-15? Please explain the rationale or reason for the design. Would the selection based on EGFP intensity, not the direct mouse ataxin-2 mRNA level, miss potential good performers since the targeted mouse ataxin-2 show sequence similarity to endogenous human ATXN2 in HEK 293T cells?

2. Besides the knocking down at the mRNA level, more experiments showing the protein level reduction of ATXN2 in vitro or in vivo would give a closer and direct correlation between ATXN2 level and TDP-43 pathology.

3. On page 8, the statement "According to our immunofluorescent analysis, we found that RfxCas13d broadly decreased the size of the YFP-TDP-43 inclusions" used the wrong figure reference (supplementary figure 3, not supplementary figure 2). What is the ATXN2 expression level in cell lines with different TDP-43 variants (TDP-43A315T, TDP-43G298S, or TDP-43M337V) overexpression? More evidence or discussion would be helpful to understand why the effect of decreasing the size of the YFP-TDP-43 variants inclusions by knocking down hATXN2 with the same crRNA and Cas13 is different.

4. Besides the therapeutic benefit in different brain regions, what's the effect of ataxin-2 mRNA targeting CRISPR platforms on the spinal cord, another TDP-43 proteinopathy-affected tissue? Although this is a proof-of-concept study, the i.v. injection dose (7 x 10¹¹ vector genomes [vg] per pup) in the in vivo experiment is much higher than the clinically safe dose. A statement or discussion on the dose and efficacy would be helpful to the translation of this study.

5. Tissue histochemistry or biochemistry toxicity analysis would be suggested to examine the in vivo safety of ataxin-2 targeting CRISPR platforms.

6. In figure 6 g and h, the targeting specificity has only cursorily been presented by examining the number of affected transcripts. The differentially expressed genes could be possibly caused by the over-expression of different Cas proteins. Is there any overlap of these differentially expressed genes? crRNA targeting sequence-dependent or not?

Minor comments:

1. In figure 2b, the representative staining images are of low quality, which may impair the accuracy of quantification.

2. In figure 3a, White stars are not obvious or informative enough to show the injection sites. A clearer cartoon or descriptions in words would help to get the experiment details better.

Reviewer #3:

Remarks to the Author:

Given that knocking down of the ataxin-2 mRNA level through RNAi could mitigate TDP-43 proteinopathy in yeast, flies and mice (Refs. 20, 26), the authors set to test the use of the RNA-targeting CRISPR systems as an alternative way to reach the same purpose.

Although the reduction of ataxin-2 mRNA level and subsequent mitigation of TDP-43 proteinopathy could be recapitulated by the authors, I do have a number of concerns about the study and the manuscript as listed in the following:

(1) As mentioned above, the knock down of the ataxin-2 mRNA to mitigate TDP-43 proteinopathies has been achieved before (actually a clinical trial is being conducted now). Thus, the CRISPR approach does not provide obvious advantage;

(2) The collateral trans-cleavage of non-target RNAs, the collateral activity (Refs. 50-54), associated with the Cas13 RNA-targeting CRISPR systems was not well addressed;

In particular, the collateral activity of the system in the experiments knocking down the endogenous ataxin-2 mRNA was checked only with use of the mCherry-encoding surrogate reporter. Suitable positive controls, as used by the authors in their transient overexpression assay as well as in previous studies of CRISPR knockdown of several other endogenous target mRNAs, were missing in these experiments;

(3) There was no data showing the knock-down efficiency of the endogenous hATXN2 by HiFi-Rfx Cas13d, which is essential for comparison of this system with the native RfxCas13d;

(4) It is difficult to draw conclusions from animal experiments involving relatively small number of animals (e.g., N=3), as the effects observed might not be statistically significant due to the high degree of animal-to-animal variability. In addition, some of the immunofluorescence staining images were not indicated with the number of replicates performed, the number of cells counted, and/or the number of tissue sections/ slides verified. Some images were also mislabeled. Furthermore, the inclusions depicted in some figures were too small for conclusive assessment of their relevance to the quantitative analysis;

(5) There was no data on the protein level of mATXN2 after the CRISPR knockdown, which is very important for the analysis of the relationship among the stress granule, the TDP-43 protein and the mATXN2 protein;

(6) On page 15:

..., including neuronal survival, the accumulation of phosphorylated and ubiquitinated TDP-43, and microgliosis and astrogliosis.

The authors counted the pTDP-43+ and ubiquitin+ inclusions separately. These data should/could not be extended to the ubiquitinated forms of TDP-43.

(7) On page 44:

Are there larger scales of brain images that could be provided in Fig. 3? On the other hand, the single HA and NeuN staining signals were way too small. The NeuN staining was missing in Fig. 3c and 3e. Also, the abbreviations were different in some of the figure legends and figures, e.g. Fig. 3 and Supplementary Fig.5, etc.;

(8) Why were some data analyzed by two-way ANOVA but others by the mixed-effect models? The authors should explain this in the method section.

Also, why for some data the authors used SD to represent the error bars, while others they used SEM?

(9) On page 47:

No representative immunofluorescence staining photos were shown in Fig. 6i and 6j. Could the authors also explain why HiFi-RfxCas13d reduced the average size of the TDP-43 A315T+ inclusions only by ~20%, when compared to the reduction of 41% by RfxCas13d?

(10) The result of Fig. 2 on the formation of TDP-43(+) granules and G3BP1(+) granules in cells under short treatment with sorbitol is different from a similar study by DeWey et al. (2011). Why?

(11) How/ Why did the authors define the TDP-43 granules in sorbitol treated cells as inclusions?

Also, what is the criteria to define soluble pTDP-43 (Fig. 5)?

Also, Western blotting data should be presented to validate the decrease of PTDP-43, as suggested by the authors from their microscopic data in Fig. 5;

POINT-BY-POINT RESPONSE

NCOMMS-23-15267

Reviewer #1:

A common feature of ALS and FTD is abnormal aggregation of TDP-43 protein in cytoplasmic inclusions. One way to reduce the toxicity associated with TDP-43 is to reduce the level of ataxin-2 protein, which is involved in the formation of stress granules. In the manuscript by Zeballos MA et al., the authors used RNA-targeting CRISPR-Cas systems to downregulate ataxin-2 expression in cellular models (HEK293T, Neuro2A) and in a mouse model of TDP-43 proteinopathy. They demonstrated successful knockdown of ataxin-2, resulting in a reduction in the number and size of TDP-43-positive inclusions and an impressive reduction in TDP-43 pathology in vivo.

Overall, this is an interesting study that confirms the role of ataxin-2 as a modifier of TDP-43 toxicity and its potential as a therapeutic target for TDP-43-related proteinopathies. The main novelty of this work is the demonstration that RNA-targeting CRISPR technology can be used to effectively downregulate ataxin-2 levels. What is missing here, however, is an expansion of our knowledge of the mechanism of action and a deeper look at the effects caused by ATXN2 silencing. Since the therapeutic strategy of silencing ataxin-2 (using ASO) in ALS patients is already in clinical trials, more mechanistic studies would be expected from subsequent works.

The manuscript is well written, the data are generally strong and well presented. There are several issues that need to be addressed.

We thank Reviewer 1 for their assessment and for noting the strengths of our study. We have revised our manuscript to address their major concerns, which altogether provide more detailed insight into the effects of silencing ATXN2 mRNA. These additions include: (i) western blot data confirming a reduction of the ataxin-2 protein in Neuro2A and HEK293T cells, the latter of which solidifies the relationship between stress granules, the TDP-43 protein and the targeting of ataxin-2, (ii) a thoroughly expanded discussion of our RNA-seq findings and (iii) new data demonstrating that targeting ataxin-2 in the motor cortex and brainstem of wild-type mice did not affect neuronal viability or lead to increases in neuroinflammation.

Major

1. Since the intermediate-length Ataxin-2 polyQ repeat expansions are significantly associated with ALS – the question remains: what was the length of the CAG repeats in the ATXN2 locus in the models used in this study? Whether the length of the polyQ domain can affect the efficiency of TDP-43 aggregates reduction and the phenotype improvement (some level of ataxin-2 still remains in the cells, polyQ expansion of Ataxin-2 increases its half-life).

The length of the polyQ repeats at the ATXN2 loci in the models used in this study is now specified in the manuscript on Pgs. 8 and 13. The mouse ataxin-2 gene does not have a polyQ expansion (ENSMUSG00000042605) while the human ATXN2 locus can carry 22-23 repeats (ENSG00000204842), sizes which we confirmed by Sanger sequencing of amplified ATXN2 sequences from Neuro2A cells, HEK293T cells and TAR4/4 mice.

We agree with Reviewer 1 on the importance for determining the effect that the polyQ repeat may have on TDP-43 aggregation and the potential consequences of this relationship on its therapeutic targeting. However, as answering this question would require the creation and/or characterization of new cell lines that express ataxin-2 variants with the intermediate-length polyQ expansion, we believe this experiment is outside the scope of the current work and better served to be thoroughly explored in a follow-up study. Nonetheless, to emphasize this important point, we have updated the Discussion on Pg. 23 to state, “**A polyQ expansion of 30-33 repeats in the ATXN2 gene is recognized as a risk factor for ALS¹⁻³. Though the mechanism(s) underlying its relationship with TDP-43 and ALS remain incompletely understood, it has been hypothesized that the polyQ expansion might make TDP-43 more susceptible to mislocalization under stress¹. Notably, the mouse and human models used in this study do not carry an intermediate-length repeat in their respective ATXN2 genes; thus, we did not determine whether RfxCas13d could suppress TDP-43 aggregation in the context of the**

polyQ expansion, an important point that will require further study, though we note that the ataxin-2-targeting ASO under evaluation in human trials is being tested in ALS patients both with and without an intermediate-length polyQ expansion (ClinicalTrials.gov Identifier: NCT04494256)."

2. The efficacy of crRNA on the endogenous mATXN2 transcript in Neuro2A cells was lower than in transfected HEK293T cells. Since we have no information about the stability of the ataxin-2 protein (half-life), WB analysis of the protein level should be performed to determine its level at a given time point.

We appreciate the opportunity to clarify this finding. Though these studies were conducted with different crRNAs (crRNA-10 for Neuro2A cells and crRNA-6 for HEK293T cells), both crRNAs exhibited roughly equivalent targeting across the two backgrounds when paired with the native RfxCas13d protein. Specifically, we previously measured a ~50% decrease in mATXN2 mRNA in Neuro2A cells and a ~40-45% decrease in hATXN2 mRNA in HEK293T cells.

Importantly, we have updated our manuscript to include measurements confirming the ability of RfxCas13d to decrease the ataxin-2 protein in both Neuro2A and HEK293T cells, results that corroborate our prior qPCR measurements. In Neuro2A cells, we find that RfxCas13d decreased the ataxin-2 protein by ~40% ($P < 0.05$; **Fig. 1d** and **Supplementary Information Fig. 2**) while, in HEK293T cells, we find that RfxCas13d decreased the ataxin-2 protein by ~42% at 48 hours post-transfection ($P < 0.05$; **Fig. 2c** and **Supplementary Information Fig. 4**), the time point at which we measured TDP-43 aggregation (**Fig. 2d-g**).

We do note that detecting the mouse ataxin-2 protein from Neuro2A cells proved challenging. We used a rabbit polyclonal anti-ataxin-2 antibody (1:500; Proteintech, 21776-1-AP) that was distinct from the antibody used in our experiments in HEK293T cells, a species cross-reactive rabbit anti-ataxin-2 polyclonal antibody from Novus Biologicals (NBP1-90063), which for us was unable to detect the full-length protein in mouse cells. All western blots are contained in the Supplementary Information in their entirety. Additionally, we show the exact values obtained for band quantification for full transparency.

3. Similarly in HEK293T cells with induced formation of stress granules the authors observed 40% reduction of ATXN2 mRNA level after 48h – what was the level of ataxin 2 protein at that time? Please mark the time points in Fig 2a. The signals are hardly visible in the photos – please improve their quality/size. Arrows showing presence or absence of inclusions are confusing. Fig. 2c-f How many cells were counted?

We thank Reviewer 1 for raising these concerns, which we have addressed. Our manuscript is now updated to include measurements of the ataxin-2 protein in HEK293T cells at 48 hours post-transfection, the exact time point at which we conducted our stress granule assays. As indicated above, we observed a ~42% decrease in the relative abundance of the protein ($P < 0.05$; **Fig. 2c** and **Supplementary Information Fig. 4**). Importantly, these findings are consistent with our measurements in the stress granule (SG) assay, which indicated a 30-40% reduction in: (i) the number of TDP-43⁺ inclusions per cell (**Fig. 2e**), (ii) the percentage cells with TDP-43⁺ inclusions (**Fig. 2f**), (iii) the average size of TDP-43⁺ inclusions per cell (**Fig. 2g**) and (iv) the percentage of cells with TDP-43⁺ and G3BPI⁺ inclusions (**Fig. 2h**). These data solidify the relationship between SG formation, the TDP-43 protein and ataxin-2.

To address the remaining concerns raised by Reviewer 1, we have substantially revised Figure 2 to: (i) denote the timepoints for SG induction, present in **Fig. 2a** and (ii) include new representative ICC images that possess improved clarity and resolution (**Fig. 2d**). Additionally, we have replaced the arrows in **Fig. 2** with new representative images that visually depict how inclusions were identified by CellProfiler.

Finally, we apologize for inadvertently omitted key information on the number of cells quantified in the analysis in **Fig. 2**. We counted >300 cells per replicate for this analysis. These details are included in both the Figure 2 Legend and in the Methods.

4. In the experiment using three TDP-43 variants with mutations in the region responsible for protein-protein interaction - what does the result mean? (especially in the context of possible interactions with ATXN2 protein?)

We appreciate the opportunity to clarify this statement. First, we apologize for any potential confusion caused by the original wording of the statement, which could be interpreted to imply that we suggested that A315T, G298S and M337V, the three TDP-43 mutations investigated in our study, reside in a domain that interacts with ataxin-2. This is likely not the case, as previous studies have indicated that TDP-43 appears to associate with ataxin-2 through its RNA recognition motifs (RRM) domains in an RNA-dependent manner¹. We aimed to state that the three TDP-43 variants tested in our study carry mutations that reside outside of the RRM domains and were thus each expected to still interact with ataxin-2 based on current knowledge. We have updated the manuscript on Pg. 10 to state, “...these three TDP-43 mutations are among the five most common ALS-linked mutations in TDP-43, **with each mutation located outside of the RNA recognition motif (RRM) domain that is believed to mediate an RNA-dependent association with the ataxin-2 protein¹. Thus, based on these insights and prior studies^{1,4}, we expected each variant to retain its ability to interact with ataxin-2.**”

This experiment therefore demonstrates that the potential effects of lowering ataxin-2 can be extended to ALS-linked TDP-43 variants, evidenced via its ability to reduce the aggregation potential of three TDP-43 variants in HEK293T cells.

5. In an in vivo experiment, the authors got spectacular results: improvement in behavioral tests, extended lifespan, reduced inflammation.... The study was done with RfxCas13d which, as they showed in Neuro2A cells, can silence other transcripts nonspecifically (over 700). Among them, about 50% were upregulated - how can this be explained? The supplementary Excel file with a list of transcripts from RNAseq is not very informative. It would be interesting to see if the most altered genes are random or clustered into some pathways? What fraction of altered transcripts are shared by RfxCas13d, HiFi-RfxCas13d and DiCas7-11?

We have revised our manuscript to more thoroughly discuss our RNA-seq findings. We now include a comparison of the enriched biological functions for each RNA-targeting CRISPR variant, as well as an analysis of the trends observed for the differentially expressed genes (DEGs) that overlapped between each variant. Unsurprisingly, for RfxCas13d, we observed clustering around themes previously linked to the function of ataxin-2⁵⁻⁷.

Beginning on Pg. 17, the manuscript now states. “**Using an over-representation analysis of gene ontology (GO) and biological process (BP) terms, we next compared the themes enriched for the differentially expressed genes (DEGs) by RfxCas13d, HiFi-RfxCas13d and DiCas7-11. For the native RfxCas13d protein, enrichment was observed for seven biological functions ($P < 0.05$; Fig. 5f and Supplementary Table 2), several of which were previously linked to ataxin-2⁵⁻⁷, including mRNA processing, mitochondrial ATP synthesis, RNA splicing and mRNA splicing via the spliceosome. We also observed an enrichment for histone H3-K4 monomethylation and histone H3-K4 dimethylation (Supplementary Table 2), indicating the possibility that depleting ataxin-2 with RfxCas13d affected transcription and/or its regulation. For HiFi-RfxCas13d, enrichment was observed for only two functions, histone H3-K4 monomethylation and histone H3-K4 dimethylation ($P < 0.05$; Fig. 5f and Supplementary Table 2). No significant enrichment ($P > 0.05$; Fig. 5f) was observed for any themes for DiCas7-11.**

Interestingly, we found that ~52%, ~79%, and ~50% of the DEGs for RfxCas13d, HiFi-RfxCas13d, and DiCas7-11, respectively, were up-regulated (Fig. 5e). Mitochondrial ATP synthesis, mRNA processing, and RNA splicing were among the biological functions enriched within the up-regulated DEG population for RfxCas13d which, as noted above, were previously linked to ataxin-2⁵⁻⁷ (Supplementary Fig. 18a and Supplementary Table 3). Histone H3-K4 monomethylation and histone H3-K4 dimethylation were the only up-regulated functions enriched for HiFi-RfxCas13d ($P < 0.05$; Supplementary Fig. 18a and Supplementary Table 3). Among the down-regulated genes, no significant themes were identified for any variant (Supplementary Fig. 18b and Supplementary Table 4).

Last, we analyzed the shared DEGs between the native RfxCas13d protein, HiFi-RfxCas13d, and DiCas7-11. In total, 67 DEGs were shared among the three variants, while 110 and 95 DEGs were shared between RfxCas13d and HiFi-RfxCas13d, and HiFi-RfxCas13d and DiCas7-11, respectively (Supplementary Fig. 19 and Supplementary Table 5). Though no significant functions ($P > 0.05$) were enriched for any shared DEGs, gene(s) related to cell adhesion, methylation, chromatin organization and transcription were found as among the most commonly shared for the variants (Supplementary Table 6).”

The results of these new analyses are present in Fig. 5f, Supplementary Fig. 18 and 19, and Supplementary Tables 2, 3, 4 and 5. Our general findings are highlighted in the Discussion on Pg. 25.

6. Ataxin 2 is an RNA-binding protein with multiple roles in RNA metabolism and we do not know the results of ATXN2 depletion. An important question that the authors should try to answer in this work is to what level can ataxin 2 be safely silenced while achieving a reduction in TDP-43 aggregate formation and its mislocalization?

We thank Reviewer 1 for highlighting this important point. To address this question, we have updated our manuscript to include new data that makes important strides toward assessing the tolerability of targeting ataxin-2 with RfxCas13d. Specifically, we have conducted an immunofluorescent analysis on sections from the motor cortex and brainstem of B6SJLF1/J mice injected with our two vectors, AAV9-CBh-RfxCas13d-mATXN2 or AAV9-CBh-RfxCas13d-NTG, to determine whether depleting ATXN2 led to neuronal loss or increased signs of neuroinflammation. This analysis was conducted on tissue from animals with a verified decrease in ATXN2 mRNA.

Within both the MC and the BS, our results show no difference in the number of NeuN⁺ cells in mice injected with AAV9-CBh-RfxCas13d-mATXN2 versus the controls ($P > 0.05$; $n = 4$ for both groups) and no change in the number of GFAP⁺ and Iba1⁺ cells in mice injected with AAV9-CBh-RfxCas13d-mATXN2 versus controls ($P > 0.05$ for both; $n = 4$ for both for each group). These results thus suggest that reducing ATXN2 mRNA by ~40-70% did not affect neuronal viability or increase the proliferation of common neuroinflammatory markers for at least the first month post-injection. These data are present in **Supplementary Fig. 11**.

Further, we have also revised our manuscript to acknowledge the need for further studies to comprehensively interrogate this matter further. In particular, the text on Pgs. 12-13 states, “**Though these results indicate that lowering mATXN2 did not affect neuronal viability or increase the proliferation of astrocytes or microglia for at least the first four-weeks post-injection in B6SJLF1/J mice, a more comprehensive and longer-term study is needed to conclusively determine the tolerability of this approach.**”

Additionally, we further contextualize these findings with the field in the Discussion on Pgs. 23-24 and state, “**The ataxin-2 protein is believed to play a prominent role in RNA metabolism. Thus, it is important to establish the tolerability of approaches that aim to lower its expression for therapeutic purposes. For this reason, we conducted an immunohistochemical analysis to determine whether targeting mATXN2 by RfxCas13d led to adverse effects in B6SJLF1/J mice. In both the MC and the BS, two areas that saw a ~70% and a ~40% decrease in mATXN2 mRNA, respectively, we measured no difference in neuronal viability and no change in the number of cells positive for GFAP and Iba1, two neuroinflammatory markers, in mice injected with AAV9-CBh-RfxCas13d-mATXN2 versus the control. Nonetheless, a more comprehensive and longer-term study will be needed to determine the tolerability of this approach and to answer whether there exists a specific threshold for safely lowering ataxin-2 and whether some region(s) in the brain and/or spinal cord might be more susceptible to potential adverse effects from its lowering. To this point, it is worth noting that ATXN2 knockout mice have been reported to show no major histological abnormalities in the brain⁸, though a proteomic analysis of such mice has revealed they can possess alterations in pathways related to the metabolism of branched-chain amino acids, fatty acids, and the citric acid cycle⁹.**”

7. Discussion section: “RNA-targeting platforms offer the theoretical advantage of reversibility” – for AAV vector-based CRISPR systems that provide stable expression over a long period of time (episomal expression), reversibility is still a matter of debate (unlike ASO).

We appreciate the recommendation by Reviewer 1. Given its speculative nature, we have removed this statement from our manuscript. This forward-looking sentence on the potential applicability of alternate CRISPR-based technologies for targeting ataxin-2 now only states, “**In addition to the gene silencing modalities discussed here, traditional CRISPR nucleases and next-generation DNA editing technologies¹⁰ could also conceivably be used to target ataxin-2.**”

Minor

1. Page 8 (at the bottom): please correct the reference to Figure S3 instead of Figure S2

We apologize for this error. The reference to **Fig. S3** (now **Supplementary Fig. 6**) has been corrected on Pg. 10.

2. Please correct Fig S3: ATXN2 instead of ATAXN2; there are no white stars in the figure

We thank the referee for identifying these errors. **Fig. S3** (now **Supplementary Fig. 6**) has been updated to correctly indicate ATXN2. We have also removed the reference to the white stars.

3. Fig. 3 I suggest adding a timeline with an overview of the study (timing of AAV injection, behavioral testing, experimental endpoints).

We appreciate the suggestion by the Reviewer. We have added a timeline, contained in both **Figures 3 and 4**, which includes an overview of the injection and measurements.

Reviewer #2:

The authors described the development and evaluation of ataxin-2 mRNA targeting CRISPR platforms (Cas13 and Cas7-11) to mitigate a TDP-43 proteinopathy. Preliminary screening and experiments were performed to identify the crRNAs knocking down ataxin-2 efficiently. The authors further demonstrated that knocking down ataxin-2 mRNA with selected crRNA and RfxCas13d attenuated the formation of TDP-43 aggregates in sorbitol-treated HEK293T cells. The therapeutic benefit of targeting ataxin-2 by RfxCas13d in a transgenic mouse model of a TDP-43 proteinopathy and specificity improvement in vitro using high-fidelity Cas13 and Cas7-11 showed the potential of RNA-targeting CRISPR technology to mitigate TDP-43 proteinopathy.

Overall, this is an exciting proof-of-concept study demonstrating the promise and significance of ataxin-2 mRNA targeting CRISPR platforms delivered by viral vector for treating TDP-43 proteinopathies, which is a complementary strategy to ASO-mediated ataxin-2 knockdown and shows some advantages on long-lasting benefits. However, several aspects need to be studied to give clearer conclusions and interpretations. It would also be helpful to address or discuss translation considerations or possible hurdles to the readers.

We thank Reviewer 2 for their feedback and for highlighting the strengths of our study. We have revised our manuscript to address their concerns, which have helped to substantially strengthen our study. Most notably, these additions include: (i) western blot data confirming a reduction of the ataxin-2 protein in Neuro2A and HEK23T cells, the latter of which solidifies the relationship between stress granules, the TDP-43 protein and the targeting of ataxin-2, (ii) measurements that demonstrate that RfxCas13d decreased ATXN2 in the spinal cord and improved neuronal viability in the same tissue, (iii) new data demonstrating that targeting ataxin-2 in the motor cortex and brainstem of wild-type mice did not affect neuronal viability or lead to increases in neuroinflammation and (iv) a thoroughly expanded discussion of our RNA-seq findings that addresses questions raised by the Reviewer, (v) evidence to suggest that the varying ability of RfxCas13d to decrease the inclusion size of different YFP-TDP-43 variants could stem from the variants starting inclusion size, and (vi) an expanded discussion of various translational considerations that must be overcome in the future, particularly related to dosage and safety.

Specific comments:

1. On page 5 (the first set of results) and in figure 1, crRNAs targeting mouse ataxin-2 mRNA were screened based on EGFP protein intensity which expressed/fused to mATXN2 by T2A peptide. Why were the ten crRNA candidates chosen within exon 5-15? Please explain the rationale or reason for the design. Would the selection

based on EGFP intensity, not the direct mouse ataxin-2 mRNA level, miss potential good performers since the targeted mouse ataxin-2 show sequence similarity to endogenous human ATXN2 in HEK 293T cells?

We thank Reviewer 2 for raising this point, which alludes to important future directions that we highlight in our **Discussion**. While computational tools capable of predicting active crRNAs for RfxCas13d now exist^{11,12}, those tools were unfortunately not available to us when we initiated this study. We instead chose to target the region corresponding to exons 5-15 in ATXN2 based on inferential observations made by our laboratory from our previous study that suggested targeting a region equivalent to this window could lead to effective knockdown.

To better elaborate on the rationale for the design of the crRNA, the text on Pgs. 5-6 now states, “We then designed ten crRNAs to target mATXN2, focusing on the region corresponding to exons 5 to 15 (Fig. 1b and Supplementary Fig. 1). **As computational methods capable of predicting active crRNAs for RfxCas13d were not available at the time we initiated this study, we based this targeting strategy on our prior results¹³, which indicated that crRNAs designed to bind a region equivalent to this window could target their transcript efficiently.**”

We chose to utilize an EGFP-based reporter to identify crRNAs owing to its compatibility with more high-throughput methods, including flow cytometry, and our past success in using it to identify active crRNAs for RfxCas13d. To better indicate this reasoning, we have updated the manuscript on Pg. 5 to state, “To facilitate the identification of crRNAs for mATXN2 in a **relatively high-throughput manner**, we created a reporter plasmid carrying the mATXN2 protein-coding sequence fused to an enhanced green fluorescence protein (EGFP) variant via a self-cleaving T2A peptide, which links mATXN2 expression to EGFP fluorescence, **thereby enabling us to assess RfxCas13d-mediated targeting by flow cytometry.**”

Reviewer 2 raises an excellent point in suggesting that relying on measurements of the ataxin-2 mRNA could enable the selection of highly effective crRNAs. We have revised our **Discussion** to note this future direction and to elaborate on how this criteria could be used in tandem with emerging computational tools. The text on Pg. 22-23 now states, “**in addition to utilizing computational tools capable of predicting active crRNAs for RfxCas13d and/or other Cas13 effectors^{11,12}, directly measuring the knockdown of the endogenous ATXN2 mRNA could prove more effective for identifying an optimal crRNA than the fluorescence-based reporter strategy used here, which we employed due to its more high-throughput nature and our previous success in using it to uncover crRNAs for RfxCas13d that functioned effectively *in vivo*¹³.**”

2. Besides the knocking down at the mRNA level, more experiments showing the protein level reduction of ATXN2 in vitro or in vivo would give a closer and direct correlation between ATXN2 level and TDP-43 pathology.

We thank Reviewer 2 for highlighting this critical point. As mentioned above to Reviewer 1, we have updated our manuscript to include measurements confirming the ability of RfxCas13d to decrease the ataxin-2 protein in both Neuro2A and HEK293T cells. Specifically, in Neuro2A cells, we find that RfxCas13d decreased the ataxin-2 protein by ~40% ($P < 0.05$; **Fig. 1d and Supplementary Information Fig. 2**) while, in HEK293T cells, we find that RfxCas13d decreased the ataxin-2 protein by ~40% at 48 hours post-transfection ($P < 0.05$; **Fig. 2c and Supplementary Information Fig. 4**), the time point at which we conducted our stress granule assay. These latter data, in particular, solidify the relationship between SG formation, the TDP-43 protein and ataxin-2. As noted above, the ~40% decrease in the ataxin-2 protein that we observed in HEK293T cells is consistent with our measurements that indicated a 30-40% reduction in: (i) the number of TDP-43⁺ inclusions per cell (**Fig. 2e**), (ii) the percentage cells with TDP-43⁺ inclusions (**Fig. 2f**), (iii) the average size of TDP-43⁺ inclusions per cell (**Fig. 2g**) and (iv) the percentage of cells with TDP-43⁺ and G3BPI⁺ inclusions (**Fig. 2h**).

3. On page 8, the statement “According to our immunofluorescent analysis, we found that RfxCas13d broadly decreased the size of the YFP-TDP-43 inclusions” used the wrong figure reference (supplementary figure 3, not supplementary figure 2). What is the ATXN2 expression level in cell lines with different TDP-43 variants (TDP-43A315T, TDP-43G298S, or TDP-43M337V) overexpression? More evidence or discussion would be helpful to

understand why the effect of decreasing the size of the YFP-TDP-43 variants inclusions by knocking down hATXN2 with the same crRNA and Cas13 is different.

We apologize for inadvertently referencing the incorrect Figure. We have updated the manuscript to correctly cite **Fig. S3** (now **Supplementary Fig. 6**). Further, we have updated the manuscript to quantify ATXN2 expression in cells transfected with the three different TDP-43 variants. According to qPCR, we measured no difference ($P > 0.05$) in relative ATXN2 mRNA in cells transfected with either TDP-43^{A315T}, TDP-43^{G298S} or TDP-43^{M337V} relative to control cells lacking a TDP-43 variant (**Supplementary Fig. 7a**). These findings thus indicate that the transfection of the various TDP-43 mutants did not influence the underlying level of ATXN2 expression.

To help determine why inclusions consisting of TDP-43^{A315T} appear to be more effectively reduced by RfxCas13d compared to TDP-43^{G298S} or TDP-43^{M337V}, we have included a new assessment of inclusion size for each TDP-43 mutant. Interestingly, our measurements revealed that HEK293T cells transfected with TDP-43^{G298S} and TDP-43^{M337V} had inclusions that were ~2.5-fold larger in size than those for TDP-43^{A315T} ($P < 0.0001$ for TDP-43^{G298S} and $P < 0.001$ for TDP-43^{M337V}). These findings thus suggest a potential relationship between inclusion size and their capacity for reduction by targeting ATXN2 using RfxCas13d. These data are presented in **Supplementary Fig. 7b**.

Altogether, the text on Pg. 10 has been updated to state, **“To investigate this difference further, we analyzed hATXN2 expression in cells expressing each TDP-43-YFP variant, finding no difference in the relative abundance of hATXN2 mRNA in cells transfected with TDP-43^{A315T}, TDP-43^{G298S} or TDP-43^{M337V} versus the controls ($P > 0.05$ for all; Supplementary Fig. 7), however, we did observe that inclusions for TDP-43^{G298S} and TDP-43^{M337V}, which were less efficiently reduced by RfxCas13d than TDP-43^{A315T}, were typically ~2.5-fold larger in size than those for TDP-43^{A315T} ($P < 0.001$ for both; Supplementary Fig. 7), indicating the size of the inclusions themselves may potentially affect the efficacy of this approach.”**

4. Besides the therapeutic benefit in different brain regions, what's the effect of ataxin-2 mRNA targeting CRISPR platforms on the spinal cord, another TDP-43 proteinopathy-affected tissue? Although this is a proof-of-concept study, the i.v. injection dose (7×10^{11} vector genomes [vg] per pup) in the in vivo experiment is much higher than the clinically safe dose. A statement or discussion on the dose and efficacy would be helpful to the translation of this study.

To address this point, we have updated our manuscript to include new studies that determine the effect of RfxCas13d in the spinal cord. Based on qPCR, we measured a ~30% reduction in mATXN2 mRNA in whole spinal cord tissue from both B6SJL/J mice and TAR4/4 injected with AAV9-CBh-RfxCas13d-mATXN2 versus the controls (**Fig. 3i** and **Fig. 4j**, $P < 0.05$; $n = 4$ for both groups in B6SJL/J mice and $n = 6$ for NTG and $n = 5$ for mATXN2 in TAR4/4 mice). No difference in hTDP-43 mRNA was observed in the spinal cord of TAR4/4 mice treated with RfxCas13d compared to the control (**Fig. 4k**; $P > 0.05$).

Further, we conducted an immunofluorescent analysis through a blinded observer to determine if RfxCas13d affected neuronal viability in TAR4/4 mice. Mice injected with AAV9-CBh-RfxCas13d-mATXN2 had ~36% more NeuN⁺ cells in the anterior horn of their lumbar spinal cord at end-stage compared to control mice (**Supplementary Fig. 14 b, c**, $P < 0.01$; $n = 4$ for both groups).

Finally, we thank Reviewer 2 for highlighting the need to discuss dosage. We have updated the Discussion to elaborate on the high-dosage used in this study, stating on Pg. 25, **“To ensure widespread delivery and efficient targeting for our proof-of-concept study, we administered RfxCas13d-encoding AAV9 vector to neonatal mice via both IV and ICV injections, as combining these routes has been found to be an effective strategy for enhancing delivery to the brain and spinal cord of neonatal mice¹⁴⁻¹⁶. In particular, we injected 7×10^{11} vg per pup and 8.4×10^{10} vg per pup via the facial vein and the lateral ventricles, respectively. We note these are high doses. Given emerging concerns related to the potential toxicity of high-dose AAV gene therapy¹⁷⁻¹⁹, it will be critical for future studies to identify the optimal dose, capsid and delivery strategy to safely and effectively target ATXN2.”**

5. Tissue histochemistry or biochemistry toxicity analysis would be suggested to examine the in vivo safety of ataxin-2 targeting CRISPR platforms.

We have conducted fluorescence tissue histochemistry to assess the tolerability of the approach. As described for Reviewer 1, we measured no difference ($P > 0.05$; $n = 4$ for both groups) in the number of NeuN⁺ cells quantified in the cortex or brainstem of mice injected with AAV9-CBh-RfxCas13d-mATXN2 versus the controls and no change in the number of GFAP⁺ and Iba1⁺ cells ($P > 0.05$ for both; $n = 4$ for both for each group) in the cortex or brainstem of mice injected with AAV9-CBh-RfxCas13d-mATXN2 versus the controls. These measurements, which were performed by CellProfiler, indicate that targeting ATXN2 did not affect neuronal viability or increase the proliferation of common neuroinflammatory markers, indicating tolerability based on these measurements. These data are contained in **Supplementary Fig. 11**.

Further, we note in both the **Results** and the **Discussion** the need for additional studies to comprehensively interrogate this matter further, particularly at longer time scales. Nonetheless, we have updated our study to include new measurements that demonstrate this approach was tolerated within the time frame of our experiments.

6. In figure 6 g and h, the targeting specificity has only cursorily been presented by examining the number of affected transcripts. The differentially expressed genes could be possibly caused by the over-expression of different Cas proteins. Is there any overlap of these differentially expressed genes? crRNA targeting sequence-dependent or not?

We thank Reviewer 2 for highlighting this point. The discussion of the differentially affected genes has been substantially expanded. Beginning on Pg. 17, the manuscript now states. **“Using an over-representation analysis of gene ontology (GO) and biological process (BP) terms, we next compared the themes enriched for the differentially expressed genes (DEGs) by RfxCas13d, HiFi-RfxCas13d and DiCas7-11. For the native RfxCas13d protein, enrichment was observed for seven biological functions ($P < 0.05$; Fig. 5f and Supplementary Table 2), several of which were previously linked to ataxin-2⁵⁻⁷, including mRNA processing, mitochondrial ATP synthesis, RNA splicing and mRNA splicing via the spliceosome. We also observed an enrichment for histone H3-K4 monomethylation and histone H3-K4 dimethylation (Supplementary Table 2), indicating the possibility that depleting ataxin-2 with RfxCas13d affected transcription and/or its regulation. For HiFi-RfxCas13d, enrichment was observed for only two functions, histone H3-K4 monomethylation and histone H3-K4 dimethylation ($P < 0.05$; Fig. 5f and Supplementary Table 2). No significant enrichment ($P > 0.05$; Fig. 5f) was observed for any themes for DiCas7-11.**

Interestingly, we found that ~52%, ~79%, and ~50% of the DEGs for RfxCas13d, HiFi-RfxCas13d, and DiCas7-11, respectively, were up-regulated (Fig. 5e). Mitochondrial ATP synthesis, mRNA processing, and RNA splicing were among the biological functions enriched within the up-regulated DEG population for RfxCas13d which, as noted above, were previously linked to ataxin-2⁵⁻⁷ (Supplementary Fig. 18a and Supplementary Table 3). Histone H3-K4 monomethylation and histone H3-K4 dimethylation were the only up-regulated functions enriched for HiFi-RfxCas13d ($P < 0.05$; Supplementary Fig. 18a and Supplementary Table 3). Among the down-regulated genes, no significant themes were identified for any variant (Supplementary Fig. 18b and Supplementary Table 4).

Last, we analyzed the shared DEGs between the native RfxCas13d protein, HiFi-RfxCas13d, and DiCas7-11. In total, 67 DEGs were shared among the three variants, while 110 and 95 DEGs were shared between RfxCas13d and HiFi-RfxCas13d, and HiFi-RfxCas13d and DiCas7-11, respectively (Supplementary Fig. 19 and Supplementary Table 5). Though no significant functions ($P > 0.05$) were enriched for any shared DEGs, gene(s) related to cell adhesion, methylation, chromatin organization and transcription were found as among the most commonly shared for the variants (Supplementary Table 6).”

To further address the comment by Reviewer 2 related to whether we may have detected DEGs possibly related to the over-expression of RfxCas13d, we note this is unlikely given the design of this experiment, which involved comparing cells transfected with RfxCas13d and crRNA-10 with those transfected with RfxCas13d and a non-

targeted crRNA. Nonetheless, our analysis demonstrate likely crRNA-dependent effects via the enrichment of biological functions previously linked to ataxin-2⁵⁻⁷.

The results of these new analyses are present in **Fig. 5f**, **Supplementary Fig. 18 and 19**, and **Supplementary Tables 2, 3, 4 and 5**. Our general findings are also highlighted in the Discussion on Pg. 25.

Minor comments:

1. In figure 2b, the representative staining images are of low quality, which may impair the accuracy of quantification.

We apologize for the low-quality images in the previous version of the manuscript. We have substantially revised **Figures 2 and 4** with higher quality representative images. In the case of **Fig. 2**, we now indicate the representative objects identified by our automated pipelines. All images analyzed by our pipeline were of the highest quality.

2. In figure 3a, White stars are not obvious or informative enough to show the injection sites. A clearer cartoon or descriptions in words would help to get the experiment details better.

We thank the Reviewer for the recommendation. We have updated **Fig. 4** with a new cartoon (**4b**) that more clearly indicates with words that ICV and IV injections were conducted in tandem.

Reviewer #3:

Given that knocking down of the ataxin-2 mRNA level through RNAi could mitigate TDP-43 proteinopathy in yeast, flies and mice (Refs. 20, 26), the authors set to test the use of the RNA-targeting CRISPR systems as an alternative way to reach the same purpose.

Although the reduction of ataxin-2 mRNA level and subsequent mitigation of TDP-43 proteinopathy could be recapitulated by the authors, I do have a number of concerns about the study and the manuscript as listed in the following:

We thank Reviewer 3 for their valuable feedback. We have revised our manuscript to address their concerns, which altogether has strengthened our manuscript. Most notably, based on their comments, we have updated our manuscript to include: (i) western blot data confirming a reduction of the ataxin-2 protein in Neuro2A and HEK293T cells, which helps to solidify the relationship between stress granules, the TDP-43 protein and the ataxin-2 protein, (ii) new positive controls that reinforce our ability to accurately detect collateral effects in Neuro2A and HEK293T cells, (iii) data that demonstrates that HiFi-RfxCas13d decreased hATXN2 mRNA in HEK293T cells, (iv) increased sample sizes for measurements involving brain and spinal cord tissue, and (v) new western blot data that shows RfxCas13d decreased pTDP-43 protein in the motor cortex of TAR4/4 mice, corroborating our prior immunofluorescent findings that indicated a reduction in its aggregates.

(1) As mentioned above, the knock down of the ataxin-2 mRNA to mitigate TDP-43 proteinopathies has been achieved before (actually a clinical trial is being conducted now). Thus, the CRISPR approach does not provide obvious advantage;

As noted by the Reviewer and indicated in our original manuscript, an antisense oligonucleotide (ASO) directed against ataxin-2 is under evaluation in a clinical trial (ClinicalTrials.gov Identifier: NCT04494256). However, despite their successes, ASOs do possess drawbacks. Among them is their transient life cycle, which can result in periods of diminished efficacy that necessitates redosing, a caveat that can also impose a physical burden on patients, as noted in our **Discussion**. By contrast, our approach involves the use of a genetically-encodable technology that can be delivered to cells by an AAV vector, which can enable the persistent expression of the encoded technology which, in the case of an RNA-targeting CRISPR protein, can result in continuous engagement. *This is a clear and obvious potential advantage.* This approach is thus an important complementary strategy to the ASO-based approaches currently under development that possesses its own advantages.

In addition, we wish to reiterate the strength of our results, which show that the AAV-based delivery of an ataxin-2-targeting CRISPR effector resulted in *dramatic* improvements in numerous functional deficits, including gait, kyphosis, tremors, weight and grip strength, survival, and several neuropathological hallmarks. These improvements compare very favorably with those observed with an ataxin-2-targeted ASO which, moving forward, illustrates the potential for a CRISPR-based approach for targeting ataxin-2.

(2) The collateral trans-cleavage of non-target RNAs, the collateral activity (Refs. 50-54), associated with the Cas13 RNA-targeting CRISPR systems was not well addressed;

In particular, the collateral activity of the system in the experiments knocking down the endogenous ataxin-2 mRNA was checked only with use of the mCherry-encoding surrogate reporter. Suitable positive controls, as used by the authors in their transient overexpression assay as well as in previous studies of CRISPR knockdown of several other endogenous target mRNAs, were missing in these experiments;

We thank Reviewer 3 for the recommendations, which we have incorporated into our manuscript. We do wish to reiterate that collateral *trans*-cleavage and targeting specificity were evaluated in our original manuscript. Using an established surrogate reporter assay²⁰⁻²⁴ we: (i) analyzed if collateral effects were induced by the native RfxCas13d protein when programmed to target both the endogenous mouse and human ATXN2 transcripts in Neuro2A and HEK293T cells, respectively, and (ii) determined the extent that collateral effects were induced by the wild-type RfxCas13d variants, its engineered high-fidelity variants and Cas7-11 when challenged to target an overexpressed mRNA transcript. We also determined the transcriptome-specificity of these variants in HEK293T cells, which provided important comparative insights.

As recommended by Reviewer 3, we have updated our manuscript to include new positive controls that confirm the accuracy of our methods for measuring collateral effects in cells. As a positive control for detecting collateral *trans*-cleavage in HEK293T cells, we utilized a validated crRNA designed to target ribosomal protein L4 (RPL4)²¹, a strongly expressed gene that, when previously targeted by the native RfxCas13d protein, triggered measurable collateral effects²¹. Using the mCherry-based surrogate reporter assay, we measured by flow cytometry a ~50% decrease in mCherry fluorescence in cells transfected with RfxCas13d and RPL4-targeting crRNA versus control cells ($P < 0.05$; **Supplementary Fig. 5**), whereas no decrease in mCherry was observed for the ATXN2-targeting crRNA.

Due to a relative lack of crRNAs verified to induce collateral effects in mouse cells at the time we initiated these experiments, we designed a new crRNA to target mouse DKK1, a strongly expressed gene that encodes the Dickkopf WNT signaling pathway inhibitor 1 (DKK1). Based on flow cytometry and relative to Neuro2A cells transfected with a non-targeted crRNA, we found that cells transfected with RfxCas13d and a mDKK1-targeting crRNA had a ~17% reduction in mCherry fluorescence intensity relative to controls ($P < 0.01$, **Supplementary Fig. 3a**) while no difference in mCherry was detected in cells transfected with RfxCas13d with a ATXN2-targeting crRNA ($P > 0.05$). qPCR confirmed that RfxCas13d targeted the DKK1 transcripts ($P < 0.01$ **Supplementary Fig. 3b**).

These new controls reaffirm the validity of our methods and support our findings indicating that targeting the ATXN2 mRNA by the native RfxCas13d protein did not induce significant collateral effects in either HEK293T cells or Neuro2A cells when targeting endogenous transcripts.

(3) There was no data showing the knock-down efficiency of the endogenous hATXN2 by HiFi-Rfx Cas13d, which is essential for comparison of this system with the native RfxCas13d;

We thank Reviewer 3 for highlighting this omission. We have updated our manuscript to include these data, which are instructive for comparing the different RfxCas13d variants used in our study. According to qPCR, we measured that RfxCas13d-N2V7, the high-fidelity variant employed in HEK293T cells, decreased hATXN2 mRNA by ~26% ($P < 0.01$, **Fig. 5h**). We now show that RfxCas13d-N2V7 decreased ATXN2 mRNA in both Neuro2A cells and HEK293T cells.

Interestingly, these results, in combination with the TDP-43⁺ inclusion assay, suggest that RfxCas13d-N2V7 is less efficient than the wild-type RfxCas13d protein in HEK293T cells when paired with crRNA-6, though both possessed an equivalent targeting ability in Neuro2A cells when paired with crRNA-10. The **Discussion** has been revised to elaborate on the apparent reduction in efficiency in this cellular background for the higher fidelity variant. We hypothesize this modest reduction could be attributed to the fact the ATXN2-targeting crRNAs were selected for their compatibility with the native RfxCas13 protein and not RfxCas13d-N2V7. The text on Pg. 22 now states, “...while we observed that HiFi-RfxCas13d could target the mATXN2 mRNA at a frequency equivalent to the native RfxCas13d protein in Neuro2A cells, the HiFi form of the enzyme was less effective in HEK293T cells, where we observed it had a decreased ability to target hATXN2. We hypothesize this could be due to a potential incompatibility between crRNA-6, the hATXN2-targeting crRNA identified in our study, and RfxCas13d-N2V7, the HiFi-RfxCas13d variant used for these measurements. Because our initial crRNA screen was performed using the native RfxCas13d protein, we expect that future optimizations or crRNA discovery efforts conducted in the context of a HiFi variant would address this concern, as subtle differences in the structure of the RfxCas13d-N2V7 protein relative to its native form could have affected its ability to effectively engage with the hATXN2 mRNA”

(4) It is difficult to draw conclusions from animal experiments involving relatively small number of animals (e.g., N=3), as the effects observed might not be statistically significant due to the high degree of animal-to-animal variability. In addition, some of the immunofluorescence staining images were not indicated with the number of replicates performed, the number of cells counted, and/or the number of tissue sections/ slides verified. Some images were also mislabeled. Furthermore, the inclusions depicted in some figures were too small for conclusive assessment of their relevance to the quantitative analysis;

To address the first point raised by Reviewer 3, we have expanded the group size of each qPCR- and immunofluorescence-based measurement from an animal tissue to at least an N = 4, with several measurements now having an N = 5 or greater. All the analyses in our manuscript, both in the originally submitted version and in this newly revised form, are statistically significant (at least P < 0.05), while the sample sizes were determined by a power analysis using an expected effect informed by the literature^{25,26}.

We apologize for omitting the various experimental details related to the quantification of our immunofluorescent analyses in the Figure Legends and thank Reviewer 3 for highlighting this oversight. We have updated each relevant Figure Legend, as well as the Methods section, to indicate the number of cells counted. We have also confirmed that all images are appropriately labeled.

(5) There was no data on the protein level of mATXN2 after the CRISPR knockdown, which is very important for the analysis of the relationship among the stress granule, the TDP-43 protein and the mATXN2 protein;

To address this important point, which was raised by each Reviewer, we have revised our manuscript to include measurements demonstrating that RfxCas13d decreased ataxin-2 protein in both Neuro2A and HEK293T cells. This includes measuring that RfxCas13d decreased the ataxin-2 protein by ~40% in HEK293T cells at 48 hours post-transfection (P < 0.05; **Fig. 2c** and **Supplementary Information Fig. 4**), the time point at which we analyzed stress granules. These data are consistent with the stress granule measurements in **Fig. 2**, which indicate a 30-40% reduction in: (i) the number of TDP-43⁺ inclusions per cell (**Fig. 2e**), (ii) the percentage cells with TDP-43⁺ inclusions (**Fig. 2f**), (iii) the average size of TDP-43⁺ inclusions per cell (**Fig. 2g**) and (iv) the percentage of cells with TDP-43⁺ and G3BPI⁺ inclusions (**Fig. 2h**).

(6) On page 15:

..., including neuronal survival, the accumulation of phosphorylated and ubiquitinated TDP-43, and microgliosis and astrogliosis.

The authors counted the pTDP-43⁺ and ubiquitin⁺ inclusions separately. These data should/could not be extended to the ubiquitinated forms of TDP-43.

The Reviewer is correct in that we separately counted pTDP-43⁺ and ubiquitin⁺ inclusions. We have carefully revised the manuscript to avoid any potential misleading statements. We now write on Pg. 20, “we observed that RfxCas13d could not only improve gait, kyphosis, tremors, weight loss, grip strength and rotarod performance, but that it could also extend lifespan and decrease the severity of several neuropathological hallmarks, including neuronal survival, the **accumulation of phosphorylated TDP-43, the deposition of ubiquitinated inclusions,** and microgliosis and astrogliosis.”

(7) On page 44:

Are there larger scales of brain images that could be provided in Fig. 3? On the other hand, the single HA and NeuN staining signals were way too small. The NeuN staining was missing in Fig. 3c and 3e. Also, the abbreviations were different in some of the figure legends and figures, e.g. Fig. 3 and Supplementary Fig.5, etc.;

We have updated **Fig. 3** to include not only large-scaler images that better illustrate the breadth of transduction by AAV, but also higher-objective insets with clearer HA and NeuN stains. We have also updated **Fig. 3c** and **Fig. 3e** to include additional NeuN staining, which demonstrate overlap with the HA epitope fused to RfxCas13d.

In order to further illustrate transduction, we have also created a new Supplementary Figure, **Supplementary Fig. 9**, which features additional images from the brainstem and spinal cord. We have also updated all abbreviations in the Figure Legends and in-text references to the Figures for consistency and correctness. We thank Reviewer 3 for identifying these points.

(8) Why were some data analyzed by two-way ANOVA but others by the mixed-effect models? The authors should explain this in the method section.

Also, why for some data the authors used SD to represent the error bars, while others they used SEM.

Due to an unequal group size, a mixed effect model was used to analyze the tremor. All other assessments were analyzed using a two-way ANOVA. The Statistical Analysis section on Pg. 39 has been updated to state, “**...gait impairment, kyphosis, grip strength and weight were compared by two-way analysis of variance (ANOVA); tremor was analyzed by a mixed effects model, which was used due to the unequal sizes of the groups for these measurements; rotarod was compared using an unpaired one-tailed *t*-test. Survival was analyzed by Kaplan-Meier analyses using the Mantel-Cox test.**” The unequal size for tremor was due to an unexpected COVID-related logistical event that impacted these measurements.

All data, except those from the behavior measurements, rely on SD for error. All behavior measurements employ SEM.

(9) On page 47:

No representative immunofluorescence staining photos were shown in Fig. 6i and 6j. Could the authors also explain why HiFi-RfxCas13d reduced the average size of the TDP-43 A315T⁺ inclusions only by ~20%, when compared to the reduction of 41% by RfxCas13d?

As requested by the Reviewer, we have added representative immunofluorescence staining images for the results in **Fig. 6i** and **Fig. 6j (updated to Fig. 5g, i in the revised version)** to the **Supplementary Information in Supplementary Fig. 20**.

As noted above, and in contrast to the measurements in Neuro2A cells, we observe by qPCR that the HiFi-RfxCas13d variant appears to target ATXN2 mRNA less efficiently in HEK293T cells compared to the native protein. We hypothesize that this less efficient targeting could help to explain its apparent reduced ability to decrease inclusions.

To elaborate on this point, we have updated the Discussion to reinforce the importance for identifying crRNAs in the context of the HiFi-RfxCas13d protein variant for the goal of achieving optimized targeting and to avoid any potential incompatibilities due to differences between the proteins. Nonetheless, our results demonstrate that HiFi-

RfxCas13d variants possess markedly improved specificity and can significantly lower ATXN2 mRNA, which altogether highlights their potential.

(10) The result of Fig. 2 on the formation of TDP-43(+) granules and G3BP1 (+) granules in cells under short treatment with sorbitol is different from a similar study by DeWey et al. (2011). Why?

We thank the Reviewer for identifying this point and providing us an opportunity to clarify. As we used G3BP1 as a marker for SGs, we have revised our manuscript to cite the most relevant and appropriate studies for our methods^{27,28}. We note that the G3BP1⁺ SGs in these studies are largely consistent in their features with those observed in our manuscript^{27,28} and were induced using a similar concentration of sorbitol (0.4 M), a similar time of exposure (1 hr) and the same cell type (HEK293T cells). The DeWey study relied on TIAR as SG marker, which could contribute to explaining any discrepancy noted by Reviewer 3.

(11) How/ Why did the authors define the TDP-43 granules in sorbitol treated cells as inclusions?

To detect inclusions within the sorbitol-treated cells, we used a similar quantification procedure as developed by Oyston et al²⁹ and Keating et al.³⁰ for quantifying TDP-43. Briefly, we first identified inclusions using the global, two-class Otsu thresholding method with a thresholding correction factor of 5 using a diameter range of 3 to 25 pixels to identify TDP-43⁺ foci and 1 to 35 pixels to identify G3BP1⁺ foci. Image segmentation of clumped objects was then determined by inclusion intensity. We then used the Relate Objects module to colocalize G3BP1⁺ foci and TDP-43⁺ foci. We have updated **Fig. 2** to more clearly depict how inclusions were detected. The details related to inclusion quantification are thoroughly described in the Methods, as well as the Figure Legends.

Also, what is the criteria to define soluble pTDP-43 (Fig. 5)?

Also, Western blotting data should be presented to validate the decrease of pTDP-43, as suggested by the authors from their microscopic data in Fig. 5;

We thank Reviewer 3 for raising these two important points, which we have addressed together. We have conducted a western blot analysis to quantify the relative abundance of pTDP-43 protein in tissue from the motor cortex of TAR4/4 mice, where we found that TAR4/4 mice injected with AAV9-CBh-RfxCas13d-mATXN2 had a ~33% reduction in pTDP-43 protein relative to the control mice ($P < 0.05$; **Fig. 4n and Supplementary Information Fig. 15**; $n = 5$ for both groups).

Given these new data and potential concerns related to quantifying a soluble protein using an immunofluorescence-based measurement, we have removed the prior soluble pTDP-43 immunofluorescent analysis and now only make the point that RfxCas13d decreased pTDP-43 protein in the MC of TAR4/4 mice.

References

1. Elden, A. C. *et al.* Ataxin-2 intermediate-length polyglutamine expansions are associated with increased risk for ALS. *Nature* **466**, 1069–1075 (2010).
2. Ross, O. A. *et al.* Ataxin-2 repeat-length variation and neurodegeneration. *Hum. Mol. Genet.* **20**, 3207–3212 (2011).
3. Gispert, S. *et al.* The modulation of Amyotrophic Lateral Sclerosis risk by Ataxin-2 intermediate polyglutamine expansions is a specific effect. *Neurobiol. Dis.* **45**, 356–361 (2012).
4. Kim, H.-J. *et al.* Therapeutic modulation of eIF2 α phosphorylation rescues TDP-43 toxicity in amyotrophic lateral sclerosis disease models. *Nat. Genet.* **46**, 152–160 (2014).
5. Yokoshi, M. *et al.* Direct Binding of Ataxin-2 to Distinct Elements in 3' UTRs Promotes mRNA Stability and Protein Expression. *Mol. Cell* **55**, 186–198 (2014).
6. Rounds, J. C. *et al.* The disease-associated proteins *Drosophila* Nab2 and Ataxin-2 interact with shared RNAs and coregulate neuronal morphology. *Genetics* **220**, iyab175 (2022).
7. Tuong Vi, D. T. *et al.* Pbp1, the yeast ortholog of human Ataxin-2, functions in the cell growth on non-fermentable carbon sources. *PLOS ONE* **16**, e0251456 (2021).
8. Kiehl, T.-R. *et al.* Generation and characterization of Sca2 (ataxin-2) knockout mice. *Biochem. Biophys. Res. Commun.* **339**, 17–24 (2006).
9. Meierhofer, D., Halbach, M., Şen, N. E., Gispert, S. & Auburger, G. Ataxin-2 (Atxn2)-knock-out mice show branched chain amino acids and fatty acids pathway alterations. *Mol. Cell. Proteomics* **15**, 1728–1739 (2016).
10. Zeballos C., M. A. & Gaj, T. Next-generation CRISPR technologies and their applications in gene and cell therapy. *Trends Biotechnol.* **39**, 692–705 (2021).
11. Wessels, H.-H. *et al.* Massively parallel Cas13 screens reveal principles for guide RNA design. *Nat. Biotechnol.* **38**, 722–727 (2020).
12. Guo, X. *et al.* Transcriptome-wide Cas13 guide RNA design for model organisms and viral RNA pathogens. *Cell Genomics* **1**, 100001 (2021).
13. Powell, J. E. *et al.* Targeted gene silencing in the nervous system with CRISPR-Cas13. *Sci. Adv.* **8**, eabk2485 (2022).
14. Di Meo, I., Marchet, S., Lamperti, C., Zeviani, M. & Viscomi, C. AAV9-based gene therapy partially ameliorates the clinical phenotype of a mouse model of Leigh syndrome. *Gene Ther.* **24**, 661–667 (2017).
15. Armbruster, N. *et al.* Efficacy and biodistribution analysis of intracerebroventricular administration of an optimized scAAV9-SMN1 vector in a mouse model of spinal muscular atrophy. *Mol. Ther. - Methods Clin. Dev.* **3**, 16060 (2016).
16. Corrà, S., Cerutti, R., Balmaceda, V., Viscomi, C. & Zeviani, M. Double administration of self-complementary AAV9 *NDUFS4* prevents Leigh disease in *Ndufs4*^{-/-} mice. *Brain* **145**, 3405–3414 (2022).
17. Morales, L., Gambhir, Y., Bennett, J. & Stedman, H. H. Broader implications of progressive liver dysfunction and lethal sepsis in two boys following systemic high-dose AAV. *Mol. Ther.* **28**, 1753–1755 (2020).
18. Palazzi, X. *et al.* Biodistribution and Tolerability of AAV-PHP.B-CBh- *SMN1* in Wistar Han Rats and Cynomolgus Macaques Reveal Different Toxicologic Profiles. *Hum. Gene Ther.* **33**, 175–187 (2022).
19. Hinderer, C. *et al.* Severe toxicity in nonhuman primates and piglets following high-dose intravenous administration of an adeno-associated virus vector expressing human SMN. *Hum. Gene Ther.* **29**, 285–298 (2018).
20. Shi, P. *et al.* Collateral activity of the CRISPR/RfxCas13d system in human cells. *Commun. Biol.* **6**, 334 (2023).
21. Tong, H. *et al.* High-fidelity Cas13 variants for targeted RNA degradation with minimal collateral effects. *Nat. Biotechnol.* (2022) doi:10.1038/s41587-022-01419-7.
22. Özcan, A. *et al.* Programmable RNA targeting with the single-protein CRISPR effector Cas7-11. *Nature* **597**, 720–725 (2021).
23. Ai, Y., Liang, D. & Wilusz, J. E. CRISPR/Cas13 effectors have differing extents of off-target effects that limit their utility in eukaryotic cells. *Nucleic Acids Res.* **50**, e65–e65 (2022).
24. Kelley, C. P., Haerle, M. C. & Wang, E. T. Negative autoregulation mitigates collateral RNase activity of repeat-targeting CRISPR-Cas13d in mammalian cells. *Cell Rep.* **40**, 111226 (2022).

25. Becker, L. A. *et al.* Therapeutic reduction of ataxin-2 extends lifespan and reduces pathology in TDP-43 mice. *Nature* **544**, 367–371 (2017).
26. Wils, H. *et al.* TDP-43 transgenic mice develop spastic paralysis and neuronal inclusions characteristic of ALS and frontotemporal lobar degeneration. *Proc. Natl. Acad. Sci.* **107**, 3858–3863 (2010).
27. Zhang, T., Baldie, G., Periz, G. & Wang, J. RNA-processing protein TDP-43 regulates FOXO-dependent protein quality control in stress response. *PLoS Genet.* **10**, e1004693 (2014).
28. Weskamp, K. *et al.* Shortened TDP43 isoforms upregulated by neuronal hyperactivity drive TDP43 pathology in ALS. *J. Clin. Invest.* **130**, 1139–1155 (2020).
29. Oyston, L. J. *et al.* Rapid in vitro quantification of TDP-43 and FUS mislocalisation for screening of gene variants implicated in frontotemporal dementia and amyotrophic lateral sclerosis. *Sci. Rep.* **11**, 14881 (2021).
30. Keating, S. S., Bademosi, A. T., San Gil, R. & Walker, A. K. Aggregation-prone TDP-43 sequesters and drives pathological transitions of free nuclear TDP-43. *Cell. Mol. Life Sci.* **80**, 95 (2023).

Reviewers' Comments:

Reviewer #1:

Remarks to the Author:

I would like to thank the authors for considering my comments and revising the manuscript accordingly.

Reviewer #2:

None

Reviewer #3:

Remarks to the Author:

In general, the authors have addressed the issues I raised with suitable responses to strengthen the manuscript.

However, although the authors successfully performed their proof-of-concept study of the mouse model through in vivo delivery using AAV vector, their approach does have potential challenges when translated for human treatment in future.

A brief note on the limitations of this approach with respect to immune-mediated toxicities should/could be added to Discussion.

POINT-BY-POINT RESPONSE

NCOMMS-23-15267A

Reviewer #1 (Remarks to the Author):

I would like to thank the authors for considering my comments and revising the manuscript accordingly.

Reviewer #3 (Remarks to the Author):

In general, the authors have addressed the issues I raised with suitable responses to strengthen the manuscript.

However, although the authors successfully performed their proof-of-concept study of the mouse model through in vivo delivery using AAV vector, their approach does have potential challenges when translated for human treatment in future.

A brief note on the limitations of this approach with respect to immune-mediated toxicities should/ could be added to Discussion.

The Discussion on Pg. 26 has been updated to elaborate on this important consideration. Our manuscript now states, **“Related to this point, another potential challenge facing the clinical implementation of AAV vectors is their immunogenicity^{94,95}. In particular, it has been well documented that immune-related responses against the AAV capsid can lower its efficacy and/or induce potential adverse events⁹⁶, a risk that can be intensified by the systemic delivery of a high-dose formulation of an AAV vector⁹⁷. To this end, the development of strategies to control and/or mitigate these effects has emerged as an important area of investigation^{95,96}.”**